# Faster and Non-ergodic $O(1/K)$ Stochastic Alternating Direction Method of Multipliers

**Cong Fang**          **Feng Cheng**          **Zhouchen Lin**[*]
Key Laboratory of Machine Perception (MOE), School of EECS, Peking University, P. R. China
Cooperative Medianet Innovation Center, Shanghai Jiao Tong University, P. R. China
fangcong@pku.edu.cn    fengcheng@pku.edu.cn    zlin@pku.edu.cn

## Abstract

We study stochastic convex optimization subjected to linear equality constraints. Traditional Stochastic Alternating Direction Method of Multipliers [1] and its Nesterov's acceleration scheme [2] can only achieve ergodic $O(1/\sqrt{K})$ convergence rates, where $K$ is the number of iteration. By introducing Variance Reduction (VR) techniques, the convergence rates improve to ergodic $O(1/K)$ [3, 4]. In this paper, we propose a new stochastic ADMM which elaborately integrates Nesterov's extrapolation and VR techniques. With Nesterov's extrapolation, our algorithm can achieve a *non-ergodic $O(1/K)$* convergence rate which is optimal for separable linearly constrained non-smooth convex problems, while the convergence rates of VR based ADMM methods are actually tight $O(1/\sqrt{K})$ in non-ergodic sense. To the best of our knowledge, this is the *first* work that achieves a truly accelerated, stochastic convergence rate for constrained convex problems. The experimental results demonstrate that our algorithm is faster than the existing state-of-the-art stochastic ADMM methods.

## 1   Introduction

We consider the following general convex finite-sum problem with linear constraints:

$$\min_{\mathbf{x}_1, \mathbf{x}_2} \quad h_1(\mathbf{x}_1) + f_1(\mathbf{x}_1) + h_2(\mathbf{x}_2) + \frac{1}{n}\sum_{i=1}^{n} f_{2,i}(\mathbf{x}_2),$$
$$s.t. \quad \mathbf{A}_1\mathbf{x}_1 + \mathbf{A}_2\mathbf{x}_2 = \mathbf{b}, \tag{1}$$

where $f_1(\mathbf{x}_1)$ and $f_{2,i}(\mathbf{x}_2)$ with $i \in \{1, 2, \cdots, n\}$ are convex and have Lipschitz continuous gradients, $h_1(\mathbf{x}_1)$ and $h_2(\mathbf{x}_2)$ are also convex, but can be non-smooth. We use the following notations: $L_1$ denotes the Lipschitz constant of $f_1(\mathbf{x}_1)$, $L_2$ is the Lipschitz constant of $f_{2,i}(\mathbf{x}_2)$ with $i \in \{1, 2, \cdots, n\}$, and $f_2(\mathbf{x}) = \frac{1}{n}\sum_{i=1}^{n} f_{2,i}(\mathbf{x})$. And we use $\nabla f$ to denote the gradient of $f$.

Problem (1) is of great importance in machine learning. The finite-sum functions $f_2(\mathbf{x}_2)$ are typically a loss over training samples, and the remaining functions control the structure or regularize the model to aid generalization [2]. The idea of using linear constraints to decouple the loss and regularization terms enables researchers to consider some more sophisticated regularization terms which might be very complicated to solve through proximity operators for Gradient Descent [5] methods. For example, for multitask learning problems [6, 7], the regularization term is set as $\mu_1\|\mathbf{x}\|_* + \mu_2\|\mathbf{x}\|_1$, for most graph-guided fused Lasso and overlapping group Lasso problem [8, 4], the regularization term can be written as $\mu\|\mathbf{A}\mathbf{x}\|_1$, and for many multi-view learning tasks [9], the regularization terms always involve $\mu_1\|\mathbf{x}\|_{2,1} + \mu_2\|\mathbf{x}\|_*$.

---

[*]Corresponding author.

Table 1: Convergence rates of ADMM type methods solving Problem (1).

| Type | Algorithm | Convergence Rate |
|---|---|---|
| Batch | ADMM [13] | Tight non-ergodic $O(\frac{1}{\sqrt{K}})$ |
| | LADM-NE [15] | Optimal non-ergodic $O(\frac{1}{K})$ |
| Stochastic | STOC-ADMM [1] | ergodic $O(\frac{1}{\sqrt{K}})$ |
| | OPG-ADMM [16] | ergodic $O(\frac{1}{\sqrt{K}})$ |
| | OPT-ADMM [2] | ergodic $O(\frac{1}{\sqrt{K}})$ |
| | SDCA-ADMM [17] | unknown |
| | SAG-ADMM [3] | Tight non-ergodic $O(\frac{1}{\sqrt{K}})$ |
| | SVRG-ADMM [4] | Tight non-ergodic $O(\frac{1}{\sqrt{K}})$ |
| | ACC-SADMM (ours) | Optimal non-ergodic $O(\frac{1}{K})$ |

Alternating Direction Method of Multipliers (ADMM) is a very popular optimization method to solve Problem (1), with its advantages in speed, easy implementation and good scalability shown in lots of literatures (see survey [10]). A popular criterion of the algorithms' convergence rate is its ergodic convergence. And it is proved in [11, 12] that ADMM converges with an $O(1/K)$ ergodic rate. However, in this paper, it is noteworthy that we consider the convergence in the non-ergodic sense. The reasons are two folded: 1) in real applications, the output of ADMM methods are non-ergodic results ($\mathbf{x}^K$), rather than the ergodic one (convex combination of $\mathbf{x}^1, \mathbf{x}^2, \cdots, \mathbf{x}^K$), as the non-ergodic results are much faster (see detailed discussions in Section 5.3); 2) The ergodic convergence rate is not trivially the same as general-case's rate. For a sequence $\{a_k\} = \{1, -1, 1, -1, 1, -1, \cdots\}$ (When $k$ is odd, $a_k$ is 1, and $-1$ when $k$ is even), it is divergent, while in ergodic sense, it converges in $O(1/K)$. So the analysis in the non-ergodic are closer to reality. 2) is especially suit for ADMM methods. In [13], Davis et al. prove that the Douglas-Rachford (DR) splitting converges in non-ergodic $O(1/\sqrt{K})$. They also construct a family of functions showing that non-ergodic $O(1/\sqrt{K})$ is tight. Chen et al. establish $O(1/\sqrt{K})$ for Linearized ADMM [14]. Then Li et al. accelerate ADMM through Nesterov's extrapolation and obtain a non-ergodic $O(1/K)$ convergence rate[15]. They also prove that the lower complexity bound of ADMM type methods for the separable linearly constrained nonsmooth convex problems is exactly $O(1/K)$, which demonstrates that their algorithm is optimal. The convergence rates for different ADMM based algorithms are shown in Table 1.

On the other hand, to meet the demands of solving large-scale machine learning problems, stochastic algorithms [18] have drawn a lot of interest in recent years. For stochastic ADMM (SADMM), the prior works are from STOC-ADMM [1] and OPG-ADMM [16]. Due to the noise of gradient, both of the two algorithms can only achieve an ergodic $O(1/\sqrt{K})$ convergence rate. There are two lines of research to accelerate SADMM. The first is to introduce the Variance Reduction (VR) [19, 20, 21] techniques into SADMM. VR methods ensure the descent direction to have a bounded variance and so can achieve faster convergence rates. The existing VR based SADMM algorithms include SDCA-ADMM [17], SAG-ADMM [3] and SVRG-ADMM [4]. SAG-ADMM and SVRG-ADMM can provably achieve ergodic $O(1/K)$ rates for Porblem (1). The second way to accelerate SADMM is through the Nesterov's acceleration [22]. This work is from [2], in which the authors propose an ergodic $O(\frac{R^2}{K^2} + \frac{D_y + \rho}{K} + \frac{\sigma}{\sqrt{K}})$ stochastic algorithm (OPT-ADMM). The dependence on the smoothness constant of the convergence rate is $O(1/K^2)$ and so each term in the convergence rate seems to have been improved to optimal. However, the worst convergence rate of it is still $O(1/\sqrt{K})$.

In this paper, we propose Accelerated Stochastic ADMM (ACC-SADMM) for large scale general convex finite-sum problems with linear constraints. By elaborately integrating Nesterov's extrapolation and VR techniques, ACC-SADMM provably achieves a non-ergodic $O(1/K)$ convergence rate which is optimal for non-smooth problems. As in non-ergodic sense, the VR based SADMM methods (e.g. SVRG-ADMM, SAG-ADMM) converges in a tight $O(1/\sqrt{K})$ (please see detailed discussions in Section 5.3), ACC-SADMM improve the convergence rates from $O(1/\sqrt{K})$ to $(1/K)$ in the ergodic sense and fill the theoretical gap between the stochastic and batch (deterministic) ADMM. The original idea to design our ACC-SADMM is by explicitly considering the snapshot vector $\tilde{\mathbf{x}}$ (approximately the mean value of $\mathbf{x}$ in the last epoch) into the extrapolation terms. This is, to some degree, inspired by [23] who proposes an $O(1/K^2)$ stochastic gradient algorithm named Katyusha for convex

Table 2: Notations and Variables

| Notation | Meaning | Variable | Meaning |
|---|---|---|---|
| $\langle \mathbf{x}, \mathbf{y} \rangle_{\mathbf{G}}, \|\mathbf{x}\|_{\mathbf{G}}$ | $\mathbf{x}^T \mathbf{G} \mathbf{y}, \sqrt{\mathbf{x}^T \mathbf{G} \mathbf{x}}$ | $\mathbf{y}_{s,1}^k, \mathbf{y}_{s,2}^k$ | extrapolation variables |
| $F_i(\mathbf{x}_i)$ | $h_i(\mathbf{x}_i) + f_i(\mathbf{x}_i)$ | $\mathbf{x}_{s,1}^k, \mathbf{x}_{s,2}^k$ | primal variables |
| $\mathbf{x}$ | $(\mathbf{x}_1, \mathbf{x}_2)$ | $\tilde{\boldsymbol{\lambda}}_s^k, \boldsymbol{\lambda}_s^k$ | dual and temp variables |
| $\mathbf{y}$ | $(\mathbf{y}_1, \mathbf{y}_2)$ | $\tilde{\mathbf{x}}_{s,1}, \tilde{\mathbf{x}}_{s,2}, \tilde{\mathbf{b}}_s$ | snapshot vectors |
| $F(\mathbf{x})$ | $F_1(\mathbf{x}_1) + F_2(\mathbf{x}_2)$ | $\mathbf{x}_1^*, \mathbf{x}_2^*, \boldsymbol{\lambda}^*$ | optimal solution of Eq. (1) |

problems. However, there are many distinctions between the two algorithms (please see detailed discussions in Section 5.1). Our method is also very efficient in practice since we have sufficiently considered the noise of gradient into our acceleration scheme. For example, we adopt extrapolation as $\mathbf{y}_s^k = \mathbf{x}_s^k + (1 - \theta_{1,s} - \theta_2)(\mathbf{x}_s^k - \mathbf{x}_s^{k-1})$ in the inner loop, where $\theta_2$ is a constant and $\theta_{1,s}$ decreases after every epoch, instead of directly adopting extrapolation as $\mathbf{y}^k = \mathbf{x}^k + \frac{\theta_1^k(1-\theta_1^{k-1})}{\theta_1^{k-1}}(\mathbf{x}^k - \mathbf{x}^{k-1})$ in the original Nesterov's scheme and adding proximal term $\frac{\|\mathbf{x}^{k+1} - \mathbf{x}^k\|^2}{\sigma k^{3/2}}$ as [2] does. There are also variants on updating of multiplier and the snapshot vector. We list the contributions of our work as follows:

- We propose ACC-SADMM for large scale convex finite-sum problems with linear constraints which integrates Nesterov's extrapolation and VR techniques. We prove that our algorithm converges in non-ergodic $O(1/K)$ which is optimal for separable linearly constrained non-smooth convex problems. To our best knowledge, this is the *first* work that achieves a truly accelerated, stochastic convergence rate for constrained convex problems.

- We do experiments on four bench-mark datasets to demonstrate the superiority of our algorithm. We also do experiments on the Multitask Learning [6] problem to demonstrate that our algorithm can be used on very large datasets.

## 2    Preliminary

Most SADMM methods alternately minimize the following variant surrogate of the augmented Lagrangian:

$$L'(\mathbf{x}_1, \mathbf{x}_2, \boldsymbol{\lambda}, \beta) = h_1(\mathbf{x}_1) + \langle \nabla f_1(\mathbf{x}_1), \mathbf{x}_1 \rangle + \frac{L_1}{2}\|\mathbf{x}_1 - \mathbf{x}_1^k\|_{\mathbf{G}_1}^2 \tag{2}$$

$$+ h_2(\mathbf{x}_2) + \langle \tilde{\nabla} f_2(\mathbf{x}_2), \mathbf{x}_2 \rangle + \frac{L_2}{2}\|\mathbf{x}_2 - \mathbf{x}_2^k\|_{\mathbf{G}_2}^2 + \frac{\beta}{2}\|\mathbf{A}_1\mathbf{x}_1 + \mathbf{A}_2\mathbf{x}_2 - \mathbf{b} + \frac{\boldsymbol{\lambda}}{\beta}\|^2,$$

where $\tilde{\nabla} f_2(\mathbf{x}_2)$ is an estimator of $\nabla f_2(\mathbf{x}_2)$ from one or a mini-batch of training samples. So the computation cost for each iteration reduces from $O(n)$ to $O(b)$ instead, where $b$ is the mini-batch size. When $f_i(\mathbf{x}) = 0$ and $\mathbf{G}_i = \mathbf{0}$, with $i = 1, 2$, Problem (1) is solved as exact ADMM. When there is no $h_i(\mathbf{x}_i)$, $\mathbf{G}_i$ is set as the identity matrix $\mathbf{I}$, with $i = 1, 2$, the subproblem in $\mathbf{x}_i$ can be solved through matrix inversion. This scheme is advocated in many SADMM methods [1, 3]. Another common approach is linearization (also called the inexact Uzawa method) [24, 25], where $\mathbf{G}_i$ is set as $\eta_i I - \frac{\beta}{L_i}\mathbf{A}_i^T\mathbf{A}_i$ with $\eta_i \geq 1 + \frac{\beta}{L_i}\|\mathbf{A}_i^T\mathbf{A}_i\|$.

For STOC-ADMM [1], $\tilde{\nabla} f_2(\mathbf{x}_2)$ is simply set as:

$$\tilde{\nabla} f_2(\mathbf{x}_2) = \frac{1}{b}\sum_{i_k \in \mathcal{I}_k} \nabla f_{2,i_k}(\mathbf{x}_2), \tag{3}$$

where $\mathcal{I}_k$ is the mini-batch of size $b$ from $\{1, 2, \cdots, n\}$. For SVRG-ADMM [4], the gradient estimator can be written as:

$$\tilde{\nabla} f_2(\mathbf{x}_2) = \frac{1}{b}\sum_{i_k \in \mathcal{I}_k} (\nabla f_{2,i_k}(\mathbf{x}_2) - \nabla f_{2,i_k}(\tilde{\mathbf{x}}_2)) + \nabla f_2(\tilde{\mathbf{x}}_2), \tag{4}$$

where $\tilde{\mathbf{x}}_2$ is a snapshot vector (mean value of last epoch).

---

**Algorithm 1** Inner loop of ACC-SADMM

---

    **for** $k = 0$ to $m - 1$ **do**

        Update dual variable: $\boldsymbol{\lambda}_s^k = \tilde{\boldsymbol{\lambda}}_s^k + \frac{\beta\theta_2}{\theta_{1,s}} \left( \mathbf{A}_1 \mathbf{x}_{s,1}^k + \mathbf{A}_2 \mathbf{x}_{s,2}^k - \tilde{\mathbf{b}}_s \right)$.

        Update $\mathbf{x}_{s,1}^{k+1}$ through Eq. (6).

        Update $\mathbf{x}_{s,2}^{k+1}$ through Eq. (7).

        Update dual variable: $\tilde{\boldsymbol{\lambda}}_s^{k+1} = \boldsymbol{\lambda}_s^k + \beta \left( \mathbf{A}_1 \mathbf{x}_{s,1}^{k+1} + \mathbf{A}_2 \mathbf{x}_{s,2}^{k+1} - \mathbf{b} \right)$.

        Update $\mathbf{y}_s^{k+1}$ through Eq. (5).

    **end for** k.

---

## 3 Our Algorithm

### 3.1 ACC-SADMM

To help readers easier understand our algorithm, we list the notations and the variables in Table 2. Our algorithm has double loops as we use SVRG [19], which also have two layers of nested loops to estimate the gradient. We denote subscript $s$ as the index of the outer loop and superscript $k$ as the index in the inner loops. For example, $\mathbf{x}_{s,1}^k$ is the value of $\mathbf{x}_1$ at the $k$-th step of the inner iteration and the $s$-th step of the outer iteration. And we use $\mathbf{x}_s^k$ and $\mathbf{y}_s^k$ to denote $(\mathbf{x}_{s,1}^k, \mathbf{x}_{s,2}^k)$, and $(\mathbf{y}_{s,1}^k, \mathbf{y}_{s,2}^k)$, respectively. In each inner loop, we update primal variables $\mathbf{x}_{s,1}^k$ and $\mathbf{x}_{s,2}^k$, extrapolation terms $\mathbf{y}_{s,1}^k$, $\mathbf{y}_{s,2}^k$ and dual variable $\boldsymbol{\lambda}_s^k$, and $s$ remains unchanged. In the outer loop, we maintain snapshot vectors $\tilde{\mathbf{x}}_{s+1,1}$, $\tilde{\mathbf{x}}_{s+1,2}$ and $\tilde{\mathbf{b}}_{s+1}$, and then assign the initial value to the extrapolation terms $\mathbf{y}_{s+1,1}^0$ and $\mathbf{y}_{s+1,2}^0$. We directly linearize both the smooth term $f_i(\mathbf{x}_i)$ and the augmented term $\frac{\beta}{2}\|\mathbf{A}_1\mathbf{x}_1 + \mathbf{A}_2\mathbf{x}_2 - \mathbf{b} + \frac{\boldsymbol{\lambda}}{\beta}\|^2$. The whole algorithm is shown in Algorithm 2.

### 3.2 Inner Loop

The inner loop of ACC-SAMM is straightforward, shown as Algorithm 1. In each iteration, we do extrapolation, and then update the primal and dual variables. There are two critical steps which ensures us to obtain a non-ergodic results. The first is extrapolation. We do extrapolation as:

$$\mathbf{y}_s^{k+1} = \mathbf{x}_s^{k+1} + (1 - \theta_{1,s} - \theta_2)(\mathbf{x}_s^{k+1} - \mathbf{x}_s^k), \tag{5}$$

We can find that $1 - \theta_{1,s} - \theta_2 \leq 1 - \theta_{1,s}$. So comparing with original Nesterov's scheme, our way is more "mild" to tackle the noise of gradient. The second step is on the updating primal variables.

$$\mathbf{x}_{s,1}^{k+1} = \underset{\mathbf{x}_1}{\arg\min}\, h_1(\mathbf{x}_1) + \langle \nabla f_1(\mathbf{y}_{s,1}^k), \mathbf{x}_1 \rangle \tag{6}$$

$$+\langle \frac{\beta}{\theta_{1,s}} \left( \mathbf{A}_1 \mathbf{y}_{s,1}^k + \mathbf{A}_2 \mathbf{y}_{s,2}^k - \mathbf{b} \right) + \boldsymbol{\lambda}_s^k, \mathbf{A}_1 \mathbf{x}_1 \rangle + \left( \frac{L_1}{2} + \frac{\beta\|\mathbf{A}_1^T\mathbf{A}_1\|}{2\theta_{1,s}} \right) \|\mathbf{x}_1 - \mathbf{y}_{s,1}^k\|^2.$$

And then update $\mathbf{x}_2$ with the latest information of $\mathbf{x}_1$, which can be written as:

$$\mathbf{x}_{s,2}^{k+1} = \underset{\mathbf{x}_2}{\arg\min}\, h_2(\mathbf{x}_2) + \langle \tilde{\nabla} f_2(\mathbf{y}_{s,1}^k), \mathbf{x}_2 \rangle + \langle \frac{\beta}{\theta_{1,s}} \left( \mathbf{A}_1 \mathbf{x}_{s,1}^{k+1} + \mathbf{A}_2 \mathbf{y}_{s,2}^k - \mathbf{b} \right) \tag{7}$$

$$+\boldsymbol{\lambda}_s^k, \mathbf{A}_2 \mathbf{x}_2 \rangle + \left( \frac{(1 + \frac{1}{b\theta_2})L_2}{2} + \frac{\beta\|\mathbf{A}_2^T\mathbf{A}_2\|}{2\theta_{1,s}} \right) \|\mathbf{x}_2 - \mathbf{y}_{s,2}^k\|^2,$$

where $\tilde{\nabla} f_2(\mathbf{y}_{s,2}^k)$ is obtained by the technique of SVRG [19] with the form:

$$\tilde{\nabla} f_2(\mathbf{y}_{s,2}^k) = \frac{1}{b} \sum_{i_{k,s} \in \mathcal{I}_{(k,s)}} \left( \nabla f_{2,i_{k,s}}(\mathbf{y}_{s,2}^k) - \nabla f_{2,i_{k,s}}(\tilde{\mathbf{x}}_{s,2}) + \nabla f_2(\tilde{\mathbf{x}}_{s,2}) \right).$$

Comparing with unaccelerated SADMM methods, which alternately minimize Eq. (2), our method is distincted in two ways. The first is that the gradient estimator are computed on the $\mathbf{y}_{s,2}^k$. The second is that we have chosen a slower increasing penalty factor $\frac{\beta}{\theta_{1,s}}$, instead of a fixed one.

**Algorithm 2** ACC-SADMM

---

**Input:** epoch length $m > 2$, $\beta$, $\tau = 2$, $c = 2$, $\mathbf{x}_0^0 = \mathbf{0}$, $\tilde{\boldsymbol{\lambda}}_0^0 = \mathbf{0}$, $\tilde{\mathbf{x}}^0 = \mathbf{x}_0^0$, $\mathbf{y}_0^0 = \mathbf{x}_0^0$,
  $\theta_{1,s} = \frac{1}{c + \tau s}$, $\theta_2 = \frac{m - \tau}{\tau(m-1)}$.

  **for** $s = 0$ to $S - 1$ **do**

    Do inner loop, as stated in Algorithm 1.

    Set primal variables: $\mathbf{x}_{s+1}^0 = \mathbf{x}_s^m$.

    Update snapshot vectors $\tilde{\mathbf{x}}_{s+1}$ through Eq. (8).

    Update dual variable: $\quad \tilde{\boldsymbol{\lambda}}_{s+1}^0 = \boldsymbol{\lambda}_s^{m-1} + \beta(1 - \tau)(\mathbf{A}_1 \mathbf{x}_{s,1}^m + \mathbf{A}_2 \mathbf{x}_{s,2}^m - \mathbf{b})$.

    Update dual snapshot variable: $\quad \tilde{\mathbf{b}}_{s+1} = \mathbf{A}_1 \tilde{\mathbf{x}}_{s+1,1} + \mathbf{A}_2 \tilde{\mathbf{x}}_{s+1,2}$.

    Update extrapolation terms $\mathbf{y}_{s+1}^0$ through Eq. (9).

  **end for** s.

**Output:** $\qquad \hat{\mathbf{x}}_S = \dfrac{1}{(m-1)(\theta_{1,S} + \theta_2) + 1} \mathbf{x}_S^m + \dfrac{\theta_{1,S} + \theta_2}{(m-1)(\theta_{1,S} + \theta_2) + 1} \sum_{k=1}^{m-1} \mathbf{x}_S^k$.

---

### 3.3 Outer Loop

The outer loop of our algorithm is a little complex, in which we preserve snapshot vectors, and then resets the initial value. The main variants we adpot is on the snapshot vector $\tilde{\mathbf{x}}_{s+1}$ and the extrapolation term $\mathbf{y}_{s+1}^0$. For the snapshot vector $\tilde{\mathbf{x}}_{s+1}$, we update it as:

$$\tilde{\mathbf{x}}_{s+1} = \frac{1}{m} \left( \left[ 1 - \frac{(\tau - 1)\theta_{1,s+1}}{\theta_2} \right] \mathbf{x}_s^m + \left[ 1 + \frac{(\tau - 1)\theta_{1,s+1}}{(m-1)\theta_2} \right] \sum_{k=1}^{m-1} \mathbf{x}_s^k \right). \qquad (8)$$

$\tilde{\mathbf{x}}_{s+1}$ is not the average of $\{\mathbf{x}_s^k\}$, different from most SVRG-based methods [19, 4]. The way of generating $\tilde{\mathbf{x}}$ guarantees a faster convergence rate for the constraints. Then we reset $\mathbf{y}_{s+1}^0$ as:

$$\mathbf{y}_{s+1}^0 = (1 - \theta_2)\mathbf{x}_s^m + \theta_2 \tilde{\mathbf{x}}_{s+1} + \frac{\theta_{1,s+1}}{\theta_{1,s}} \left[ (1 - \theta_{1,s})\mathbf{x}_s^m - (1 - \theta_{1,s} - \theta_2)\mathbf{x}_s^{m-1} - \theta_2 \tilde{\mathbf{x}}_s \right]. \qquad (9)$$

## 4 Convergence Analysis

In this section, we give the convergence results of ACC-SADMM. The proof and a outline can be found in Supplementary Material. As we have mentioned in Section 3.2, the main strategy that enable us to obtain a non-ergodic results is that we adopt extrapolation as Eq. (5). We first analyze each inner iteration, shown in Lemma 1. We ignore subscript $s$ as $s$ is unchanged in the inner iteration.

**Lemma 1** *Assume that $f_1(\mathbf{x}_1)$ and $f_{2,i}(\mathbf{x}_2)$ with $i \in \{1, 2, \cdots, n\}$ are convex and have Lipschitz continuous gradients. $L_1$ is the Lipschitz constant of $f_1(\mathbf{x}_1)$. $L_2$ is the Lipschitz constant of $f_{2,i}(\mathbf{x}_2)$ with $i \in \{1, 2, \cdots, n\}$. $h_1(\mathbf{x}_1)$ and $h_2(\mathbf{x}_2)$ is also convex. For Algorithm 2, in any epoch, we have*

$$\mathbb{E}_{i_k} \left[ L(\mathbf{x}_1^{k+1}, \mathbf{x}_2^{k+1}, \boldsymbol{\lambda}^*) \right] - \theta_2 L(\tilde{\mathbf{x}}_1, \tilde{\mathbf{x}}_2, \boldsymbol{\lambda}^*) - (1 - \theta_2 - \theta_1)L(\mathbf{x}_1^k, \mathbf{x}_2^k, \boldsymbol{\lambda}^*)$$

$$\leq \quad \frac{\theta_1}{2\beta} \left( \|\hat{\boldsymbol{\lambda}}^k - \boldsymbol{\lambda}^*\|^2 - \mathbb{E}_{i_k} \left[ \|\hat{\boldsymbol{\lambda}}^{k+1} - \boldsymbol{\lambda}^*\|^2 \right] \right) + \frac{1}{2} \|\mathbf{y}_1^k - (1 - \theta_1 - \theta_2)\mathbf{x}_1^k - \theta_2 \tilde{\mathbf{x}}_1 - \theta_1 \mathbf{x}_1^*\|_{\mathbf{G}_3}^2$$

$$- \frac{1}{2} \mathbb{E}_{i_k} \left( \|\mathbf{x}_1^{k+1} - (1 - \theta_1 - \theta_2)\mathbf{x}_1^k - \theta_2 \tilde{\mathbf{x}}_1 - \theta_1 \mathbf{x}_1^*\|_{\mathbf{G}_3}^2 \right)$$

$$+ \frac{1}{2} \|\mathbf{y}_2^k - (1 - \theta_1 - \theta_2)\mathbf{x}_2^k - \theta_2 \tilde{\mathbf{x}}_2 - \theta_1 \mathbf{x}_2^*\|_{\mathbf{G}_4}^2$$

$$- \frac{1}{2} \mathbb{E}_{i_k} \left( \|\mathbf{x}_2^{k+1} - (1 - \theta_1 - \theta_2)\mathbf{x}_2^k - \theta_2 \tilde{\mathbf{x}}_2 - \theta_1 \mathbf{x}_2^*\|_{\mathbf{G}_4}^2 \right),$$

*where $\mathbb{E}_{i_k}$ denotes that the expectation is taken over the random samples in the minibatch $\mathcal{I}_{k,s}$, $L(\mathbf{x}_1, \mathbf{x}_2, \boldsymbol{\lambda}) = F_1(\mathbf{x}_1) + F_2(\mathbf{x}_2) + \langle \boldsymbol{\lambda}, \mathbf{A}_1 \mathbf{x}_1 + \mathbf{A}_2 \mathbf{x}_2 - \mathbf{b} \rangle$ and $\hat{\boldsymbol{\lambda}}^k = \tilde{\boldsymbol{\lambda}}^k + \frac{\beta(1 - \theta_1)}{\theta_1}(\mathbf{A}\mathbf{x}^k - \mathbf{b})$, $\mathbf{G}_3 = \left( L_1 + \frac{\beta \|\mathbf{A}_1^T \mathbf{A}_1\|}{\theta_1} \right) \mathbf{I} - \frac{\beta \mathbf{A}_1^T \mathbf{A}_1}{\theta_1}$, and $\mathbf{G}_4 = \left( (1 + \frac{1}{b\theta_2})L_2 + \frac{\beta \|\mathbf{A}_2^T \mathbf{A}_2\|}{\theta_1} \right) \mathbf{I}$.*

Then Theorem 1 analyses ACC-SADMM in the whole iteration, which is the key convergence result of the paper.

**Theorem 1** *If the conditions in Lemma 1 hold, then we have*

$$\mathbb{E}\left(\frac{1}{2\beta}\|\frac{\beta m}{\theta_{1,S}}\left(\mathbf{A}\hat{\mathbf{x}}_S-\mathbf{b}\right)-\frac{\beta(m-1)\theta_2}{\theta_{1,0}}\left(\mathbf{A}\mathbf{x}_0^0-\mathbf{b}\right)+\tilde{\boldsymbol{\lambda}}_0^0-\boldsymbol{\lambda}^*\|^2\right) \quad (10)$$

$$+\mathbb{E}\left(\frac{m}{\theta_{1,S}}\left(F(\hat{\mathbf{x}}_S)-F(\mathbf{x}^*)+\langle\boldsymbol{\lambda}^*,\mathbf{A}\hat{\mathbf{x}}_S-\mathbf{b}\rangle\right)\right)$$

$$\leq \quad C_3\left(F(\mathbf{x}_0^0)-F(\mathbf{x}^*)+\langle\boldsymbol{\lambda}^*,\mathbf{A}\mathbf{x}_0^0-\mathbf{b}\rangle\right)+\frac{1}{2\beta}\|\tilde{\boldsymbol{\lambda}}_0^0+\frac{\beta(1-\theta_{1,0})}{\theta_{1,0}}(\mathbf{A}\mathbf{x}_0^0-\mathbf{b})-\boldsymbol{\lambda}^*\|^2$$

$$+\frac{1}{2}\|\mathbf{x}_{0,1}^0-\mathbf{x}_1^*\|^2_{\left(\theta_{1,0}L_1+\|\mathbf{A}_1^T\mathbf{A}_1\|\right)\mathbf{I}-\mathbf{A}_1^T\mathbf{A}_1}+\frac{1}{2}\|\mathbf{x}_{0,2}^0-\mathbf{x}_2^*\|^2_{\left((1+\frac{1}{b\theta_2})\theta_{1,0}L_2+\|\mathbf{A}_2^T\mathbf{A}_2\|\right)\mathbf{I}},$$

*where $C_3=\frac{1-\theta_{1,0}+(m-1)\theta_2}{\theta_{1,0}}$.*

Corollary 1 directly demonstrates that ACC-SADMM have a non-ergodic $O(1/K)$ convergence rate.

**Corollary 1** *If the conditions in Lemma 1 holds, we have*

$$\mathbb{E}|F(\hat{\mathbf{x}}_S)-F(\mathbf{x}^*)| \leq O(\frac{1}{S}),$$

$$\mathbb{E}\|\mathbf{A}\hat{\mathbf{x}}_S-\mathbf{b}\| \leq O(\frac{1}{S}). \quad (11)$$

We can find that $\hat{\mathbf{x}}_S$ depends on the latest $m$ information of $\mathbf{x}_s^k$. So our convergence results is in non-ergodic sense, while the analysis for SVRG-ADMM [4] and SAG-ADMM [3] is in ergodic sense, since they consider the point $\hat{\mathbf{x}}_S=\frac{1}{mS}\sum_{s=1}^S\sum_{k=1}^m\mathbf{x}_s^k$, which is the convex combination of $\mathbf{x}_s^k$ over *all* the iterations.

Now we directly use the theoretical results of [15] to demonstrate that our algorithm is optimal when there exists non-smooth term in the objective function.

**Theorem 2** *For the following problem:*

$$\min_{\mathbf{x}_1,\mathbf{x}_2} F_1(\mathbf{x}_1)+F_2(\mathbf{x}_2),\ \ s.t.\ \mathbf{x}_1-\mathbf{x}_2=\mathbf{0}, \quad (12)$$

*let the ADMM type algorithm to solve it be:*

- *Generate $\boldsymbol{\lambda}_2^k$ and $\mathbf{y}_2^k$ in any way,*

- $\mathbf{x}_1^{k+1}=Prox_{F_1/\beta^k}\left(\mathbf{y}_2^k-\frac{\boldsymbol{\lambda}_2^k}{\beta^k}\right),$

- *Generate $\boldsymbol{\lambda}_1^{k+1}$ and $\mathbf{y}_1^{k+1}$ in any way,*

- $\mathbf{x}_2^{k+1}=Prox_{F_2/\beta^k}\left(\mathbf{y}_1^{k+1}-\frac{\boldsymbol{\lambda}_1^{k+1}}{\beta^k}\right).$

*Then there exist convex functions $F_1$ and $F_2$ defined on $\mathcal{X}=\{\mathbf{x}\in\mathcal{R}^{6k+5}:\|\mathbf{x}\|\leq B\}$ for the above general ADMM method, satsifying*

$$L\|\hat{\mathbf{x}}_2^k-\hat{\mathbf{x}}_1^k\|+|F_1(\hat{\mathbf{x}}_1^k)-F_1(\mathbf{x}_1^*)+F_1(\hat{\mathbf{x}}_2^k)-F_2(\mathbf{x}_2^*)|\geq\frac{LB}{8(k+1)}, \quad (13)$$

*where $\hat{\mathbf{x}}_1^k=\sum_{i=1}^k\alpha_1^i\mathbf{x}_1^i$ and $\hat{\mathbf{x}}_2^k=\sum_{i=1}^k\alpha_2^i\mathbf{x}_2^i$ for any $\alpha_1^i$ and $\alpha_2^i$ with $i$ from 1 to k.*

Theorem 2 is Theorem 11 in [15]. More details can be found in it. Problem (12) is a special case of Problem (1) as we can set each $F_{2,i}(\mathbf{x}_2)=F(\mathbf{x}_2)$ with $i=1,\cdots,n$ or set $n=1$. So there is no better ADMM type algorithm which converges faster than $O(1/K)$ for Problem (1).

## 5 Discussions

We discuss some properties of ACC-SADMM and make further comparisons with some related methods.

Table 3: Size of datasets and mini-batch size we adopt in the experiments

| Problem | Dataset | # training | # testing | # dimension × # class | # minibatch |
|---|---|---|---|---|---|
| Lasso | a9a | $72,876$ | $72,875$ | $74 \times 2$ | $100$ |
| | covertype | $290,506$ | $290,506$ | $54 \times 2$ | |
| | mnist | $60,000$ | $10,000$ | $784 \times 10$ | |
| | dna | $2,400,000$ | $600,000$ | $800 \times 2$ | $500$ |
| Multitask | ImageNet | $1,281,167$ | $50,000$ | $4,096 \times 1,000$ | $2,000$ |

## 5.1 Comparison with Katyusha

As we have mentioned in Introduction, some intuitions of our algorithm are inspired by Katyusha [23], which obtains an $O(1/K^2)$ algorithm for convex finite-sum problems. However, Katyusha cannot solve the problem with linear constraints. Besides, Katyusha uses the Nesterov's second scheme to accelerate the algorithm while our method conducts acceleration through Nesterov's extrapolation (Nesterov's first scheme). And our proof uses the technique of [26], which is different from [23]. Our algorithm can be easily extended to unconstrained convex finite-sum and can also obtain a $O(1/K^2)$ rate but belongs to the Nesterov's first scheme [2].

## 5.2 The Growth of Penalty Factor $\frac{\beta}{\theta_{1,s}}$

The penalty factor $\frac{\beta}{\theta_{1,s}}$ increases linearly with the iteration. One might deem that this make our algorithm impractical because after dozens of epoches, the large value of penalty factor might slow down the decrement of function value. However, we have not found any bad influence. There may be two reasons 1. In our algorithm, $\theta_{1,s}$ decreases after each epoch ($m$ iterations), which is much slower than LADM-NE [15]. So the growth of penalty factor works as a continuation technique [28], which may help to decrease the function value. 2. From Theorem 1, our algorithm converges in $O(1/S)$ whenever $\theta_{1,s}$ is large. So from the theoretical viewpoint, a large $\theta_{1,s}$ cannot slow down our algorithm. We find that OPT-ADMM [2] also needs to decrease the step size with the iteration. However, its step size decreasing rate is $O(k^{\frac{3}{2}})$ and is faster than ours.

## 5.3 The Importance of Non-ergodic $O(1/K)$

SAG-ADMM [3] and SVRG-ADMM [4] accelerate SADMM to ergodic $O(1/K)$. In Theorem 9 of [15], the authors generate a class of functions showing that the original ADMM has a tight non-ergodic $O(1/\sqrt{K})$ convergence rate. When $n = 1$, SAG-ADMM and SVRG-ADMM are the same as batch ADMM, so their convergence rates are no better than $O(1/\sqrt{K})$. So in non-ergodic sense, our algorithm does have a faster convergence rate than VR based SADMM methods.

Then we are to highlight the importance of our non-ergodic result. As we have mentioned in the Introduction, in practice, the output of ADMM methods is the non-ergodic result $x^K$, not the mean of $x^1$ to $x^K$. For deterministic ADMM, the proof of ergodic $O(1/K)$ rate is proposed in [11], after ADMM had become a prevailing method of solving machine learning problems [29]; for stochastic ADMM, e.g. SVRG-ADMM [4], the authors give an ergodic $O(1/K)$ proof, but in experiment, what they emphasize to use is the mean value of the last epoch as the result. As the non-ergodic results are more close to reality, our algorithm is much faster than VR based SADMM methods, even when its rate is seemingly the same. Actually, though VR based SADMM methods have provably faster rates than STOC-ADMM, the improvement in practice is evident only after numbers of iterations, when point are close to the convergence point, rather than at the early stage. In both [3] and [4], the authors claim that SAG-ADMM and SVRG-ADMM are sensitive to initial points. We also find that if the step sizes are set based on the their theoretical guidances, sometimes they are even slower than STOC-ADMM (see Fig. 1) as the early stage lasts longer when the step size is small. Our algorithm is faster than the two algorithms which demonstrates that Nesterov's extrapolation has truly accelerated the speed and the integration of extrapolation and VR techniques is harmonious and complementary.

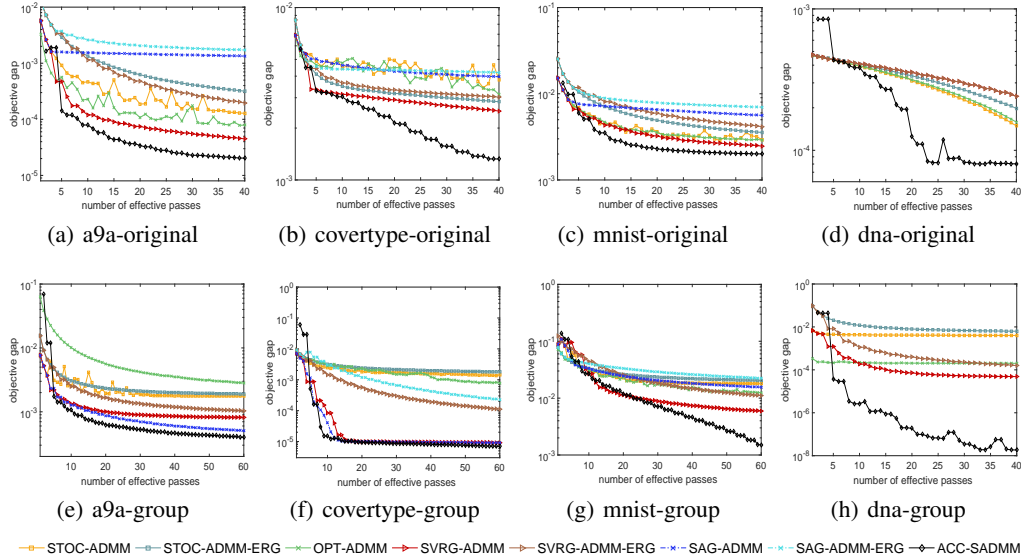

| STOC-ADMM | STOC-ADMM-ERG | OPT-ADMM | SVRG-ADMM | SVRG-ADMM-ERG | SAG-ADMM | SAG-ADMM-ERG | ACC-SADMM |

Figure 1: Experimental results of solving the original Lasso (Top) and Graph-Guided Fused Lasso (Bottom). The computation time includes the cost of calculating full gradients for SVRG based methods. SVRG-ADMM and SAG-ADMM are initialized by running STOC-ADMM for $\frac{3n}{b}$ iterations. "-ERG" represents the ergodic results for the corresponding algorithms.

## 6 Experiments

We conduct experiments to show the effectiveness of our method[3]. We compare our method with the following the-state-of-the-art SADMM algorithms: (1) STOC-ADMM [1], (2) SVRG-ADMM [4], (3) OPT-SADMM [2], (4) SAG-ADMM [3]. We ignore SDCA-ADMM [17] in our comparison since it gives no analysis on general convex problems and it is also not faster than SVRG-ADMM [4]. Experiments are performed on Intel(R) CPU i7-4770 @ 3.40GHz machine with 16 GB memory. Our experiments focus on two typical problems [4]: Lasso Problem and Multitask Learning. Due to space limited, the experiment of Multitask Learning is shown in Supplementary Materials. For the Lasso problems, we perform experiments under the following typical variations. The first is the original Lasso problem; and the second is Graph-Guided Fused Lasso model: $\min_{\mathbf{x}} \mu \|\mathbf{A}\mathbf{x}\|_1 + \frac{1}{n} \sum_{i=1}^{n} l_i(\mathbf{x})$, where $l_i(\mathbf{x})$ is the logistic loss on sample $i$, and $\mathbf{A} = [\mathbf{G}; \mathbf{I}]$ is a matrix encoding the feature sparsity pattern. $\mathbf{G}$ is the sparsity pattern of the graph obtained by sparse inverse covariance estimation [30]. The experiments are performed on four benchmark data sets: a9a, covertype, mnist and dna[4]. The details of the dataset and the mini-batch size that we use in all SADMM are shown in Table 3. And like [3] and [4], we fix $\mu = 10^{-5}$ and report the performance based on $(\mathbf{x}_t, \mathbf{A}\mathbf{x}_t)$ to satisfy the constraints of ADMM. Results are averaged over five repetitions. And we set $m = \frac{2n}{b}$ for all the algorithms. For original Lasso problem, the step sizes are set through theoretical guidances for each algorithm. For the Graph-Guided Lasso, the best step sizes are obtained through searches on parameters which give best convergence progress. Except ACC-SADMM, we use the continuation technique [28] to accelerate algorithms. SAG-ADMM is performed on the first three datasets due to its large memory requirement.

The experimental results are shown in Fig. 1. We can find that our algorithm consistently outperforms other compared methods in all these datasets for both the two problems, which verifies our theoretical analysis. The details about parameter setting, experimental results where we set a larger fixed step size for the group guided Lasso problem, curves of the test error, the memory costs of all algorithms, and Multitask learning experiment are shown in Supplementary Materials.

# 7  Conclusion

We propose ACC-SADMM for the general convex finite-sum problems. ACC-SADMM integrates Nesterov's extrapolation and VR techniques and achieves a non-ergodic $O(1/K)$ convergence rate, which shows theoretical and practical importance. We do experiments to demonstrate that our algorithm is faster than other SADMM methods.

## Acknowledgment

Zhouchen Lin is supported by National Basic Research Program of China (973 Program) (grant no. 2015CB352502) and National Natural Science Foundation (NSF) of China (grant no.s 61625301, 61731018, and 61231002).

## Footnotes

[2]We follow [26] to name the extrapolation scheme as Nesterov's first scheme and the three-step scheme [27] as the Nesterov's second scheme.

[3]The code will be available at `http://www.cis.pku.edu.cn/faculty/vision/zlin/zlin.htm`.

[4]a9a, covertype and dna are from: `http://www.csie.ntu.edu.tw/~cjlin/libsvmtools/datasets/`, and mnist is from: `http://yann.lecun.com/exdb/mnist/`.

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
