[Supplementary Material · Supplementary Material of ACCSADMM.pdf]

# Supplementary Material of
# Faster and Non-ergodic $O(1/K)$ Stochastic Alternating Direction Method of Multipliers

Cong Fang     Feng Cheng     Zhouchen Lin[*]

Key Laboratory of Machine Perception (MOE), School of EECS, Peking University

Cooperative Medianet Innovation Center, Shanghai Jiao Tong University

fangcong@pku.edu.cn    fengcheng@pku.edu.cn    zlin@pku.edu.cn

The Supplementary Material is structured as follows: We give a outline of the proof in Section 1. In Section 2, we proof Lemma 1, Theorem 1, and Corollary 1 in the paper. In Section 3, we demonstrate the details and more results of the experiments.

## 1   Outline of Proof

Below is the outline of our proof. We will ignore the subscript $s$ in the proof of Step 1, Step 2, Eq. (3) and Eq. (4) in Step 3, and Step 4, since the analysis are in a single epoch and $s$ is fixed in these steps.

Step: 1
We analyze $\mathbf{x}_1$. Through the optimal solution of $\mathbf{x}_1$ in Eq (6) of the paper, and the convexity of $F_1(\cdot)$, we can obtain:

$$
\begin{aligned}
& F_1(\mathbf{x}_1^{k+1}) \\
\leq \quad & (1 - \theta_1 - \theta_2) F_1(\mathbf{x}_1^k) + \theta_2 F_1(\tilde{\mathbf{x}}_1) + \theta_1 F_1(\mathbf{x}_1^*) \\
& - \langle \mathbf{A}_1^T \bar{\boldsymbol{\lambda}}(\mathbf{x}_1^{k+1}, \mathbf{y}_2^k), \mathbf{x}_1^{k+1} - (1 - \theta_1 - \theta_2)\mathbf{x}_1^k - \theta_2 \tilde{\mathbf{x}}_1 - \theta_1 \mathbf{x}_1^* \rangle + \frac{L_1}{2} \|\mathbf{x}_1^{k+1} - \mathbf{y}_1^k\|^2 \\
& - \left\langle \mathbf{x}_1^{k+1} - \mathbf{y}_1^k, \mathbf{x}_1^{k+1} - (1 - \theta_1 - \theta_2)\mathbf{x}_1^k - \theta_2 \tilde{\mathbf{x}}_1 - \theta_1 \mathbf{x}^* \right\rangle_{\left( L_1 + \frac{\beta \|\mathbf{A}_1^T \mathbf{A}_1\|}{\theta_1} \right) \mathbf{I} - \frac{\beta \mathbf{A}_1^T \mathbf{A}_1}{\theta_1}},
\end{aligned}
\tag{1}
$$

where we set $\bar{\boldsymbol{\lambda}}(\mathbf{x}_1, \mathbf{x}_2) = \frac{\beta}{\theta_1}(\mathbf{A}_1 \mathbf{x}_1 + \mathbf{A}_2 \mathbf{x}_2 - \mathbf{b}) + \boldsymbol{\lambda}^k$.

Step: 2
We analyze $\mathbf{x}_2$. Through the optimal solution of $\mathbf{x}_2$ in Eq (7) of the paper, and

---

[*]Corresponding author.

the convexity of $F_2(\cdot)$, we can obtain:

$$
\begin{aligned}
&\mathbb{E}_{i_k} F_2(\mathbf{x}_2^{k+1}) \\
\leq\ &-\mathbb{E}_{i_k} \left\langle \mathbf{A}_2^T \bar{\boldsymbol{\lambda}}(\mathbf{x}_1^{k+1}, \mathbf{y}_2^k) + \left( \alpha L_2 + \frac{\beta \|\mathbf{A}_2^T \mathbf{A}_2\|}{\theta_1} \right) (\mathbf{x}_2^{k+1} - \mathbf{y}_2^k), \mathbf{x}_2^{k+1} - \theta_2 \tilde{\mathbf{x}}_2 \right\rangle \\
&-\mathbb{E}_{i_k} \left\langle \mathbf{A}_2^T \bar{\boldsymbol{\lambda}}(\mathbf{x}_1^{k+1}, \mathbf{y}_2^k) + \left( \alpha L_2 + \frac{\beta \|\mathbf{A}_2^T \mathbf{A}_2\|}{\theta_1} \right) (\mathbf{x}_2^{k+1} - \mathbf{y}_2^k), -(1 - \theta_2 - \theta_1)\mathbf{x}_2^k - \theta_1 \mathbf{x}_2^* \right\rangle \\
&+(1 - \theta_2 - \theta_1) F_2(\mathbf{x}_2^k) + \theta_1 F_2(\mathbf{x}_2^*) + \theta_2 F_2(\tilde{\mathbf{x}}_2) + \mathbb{E}_{i_k} \left( \frac{(1 + \frac{1}{b\theta_2}) L_2}{2} \|\mathbf{x}_2^{k+1} - \mathbf{y}_2^k\|^2 \right), \quad (2)
\end{aligned}
$$

where $\mathbb{E}_{i_k}$ indicates that the expectation is taken over the random samples in the minibatch $\mathcal{I}_{k,s}$, under the condition that $\mathbf{y}_2^k$, $\tilde{\mathbf{x}}_2$ and $\mathbf{x}_2^k$ (the randomness in the first $sm+k$ iterations are fixed) are known and $\alpha = \frac{1}{b\theta_2}$. In step 2, we study the point at $\mathbf{w}^k = \mathbf{y}_2^k + \theta_3(\mathbf{y}_2^k - \tilde{\mathbf{x}}_2)$ and $\mathbf{z}^{k+1} = \mathbf{x}_2^{k+1} + \theta_3(\mathbf{y}_2^k - \tilde{\mathbf{x}}_2)$, where $\theta_3$ is an undetermined coefficient, which helps us eliminate the effect of the variance in the stochastic gradient.

Step: 3
We consider the multiplier. Setting $\hat{\boldsymbol{\lambda}}^k = \tilde{\boldsymbol{\lambda}}^k + \frac{\beta(1-\theta_1)}{\theta_1}(\mathbf{A}_1 \mathbf{x}_1^k + \mathbf{A}_2 \mathbf{x}_2^k - \mathbf{b})$, it has the following properties:

$$
\begin{aligned}
\hat{\boldsymbol{\lambda}}^{k+1} &= \bar{\boldsymbol{\lambda}}(\mathbf{x}_1^{k+1}, \mathbf{x}_2^{k+1}), && (3) \\
\hat{\boldsymbol{\lambda}}^{k+1} - \hat{\boldsymbol{\lambda}}^k &= \frac{\beta A_1}{\theta_1} \left( \mathbf{x}_1^{k+1} - (1-\theta_1)\mathbf{x}_1^k - \theta_1 \mathbf{x}_1^* + \theta_2(\mathbf{x}_1^k - \tilde{\mathbf{x}}_1) \right), \\
&\quad + \frac{\beta A_2}{\theta_1} \left( \mathbf{x}_2^{k+1} - (1-\theta_1)\mathbf{x}_2^k - \theta_1 \mathbf{x}_2^* + \theta_2(\mathbf{x}_2^k - \tilde{\mathbf{x}}_2) \right), && (4) \\
\hat{\boldsymbol{\lambda}}_s^0 &= \hat{\boldsymbol{\lambda}}_{s-1}^m, \quad s \geq 1. && (5)
\end{aligned}
$$

Step: 4
Define $L(\mathbf{x}_1, \mathbf{x}_2, \boldsymbol{\lambda}) = F_1(\mathbf{x}_1) - F_1(\mathbf{x}_1^*) + F_2(\mathbf{x}_2) - F_2(\mathbf{x}_2^*) + \langle \boldsymbol{\lambda}, \mathbf{A}_1 \mathbf{x}_1 + \mathbf{A}_2 \mathbf{x}_2 - \mathbf{b} \rangle$.

Adding Eq. (1) and Eq. (2), and simplifying the result, we obtain Lemma 1:

$$\mathbb{E}_{i_k}\left(L(\mathbf{x}_1^{k+1}, \mathbf{x}_2^{k+1}, \boldsymbol{\lambda}^*)\right) - \theta_2 L(\tilde{\mathbf{x}}_1, \tilde{\mathbf{x}}_2, \boldsymbol{\lambda}^*) - (1 - \theta_2 - \theta_1)L(\mathbf{x}_1^k, \mathbf{x}_2^k, \boldsymbol{\lambda}^*) \qquad (6)$$

$$\leq \frac{\theta_1}{2\beta}\left(\|\hat{\boldsymbol{\lambda}}^k - \boldsymbol{\lambda}^*\|^2 - \mathbb{E}_{i_k}\|\hat{\boldsymbol{\lambda}}^{k+1} - \boldsymbol{\lambda}^*\|^2\right)$$

$$+ \frac{1}{2}\|\mathbf{y}_1^k - (1 - \theta_1 - \theta_2)\mathbf{x}_1^k - \theta_2\tilde{\mathbf{x}}_1 - \theta_1\mathbf{x}_1^*\|^2_{\left(L_1 + \frac{\beta\|\mathbf{A}_1^T\mathbf{A}_1\|}{\theta_1}\right)\mathbf{I} - \frac{\beta\mathbf{A}_1^T\mathbf{A}_1}{\theta_1}}$$

$$- \frac{1}{2}\mathbb{E}_{i_k}\left(\|\mathbf{x}_1^{k+1} - (1 - \theta_1 - \theta_2)\mathbf{x}_1^k - \theta_2\tilde{\mathbf{x}}_1 - \theta_1\mathbf{x}_1^*\|^2_{\left(L_1 + \frac{\beta\|\mathbf{A}_1^T\mathbf{A}_1\|}{\theta_1}\right)\mathbf{I} - \frac{\beta\mathbf{A}_1^T\mathbf{A}_1}{\theta_1}}\right)$$

$$+ \frac{1}{2}\|\mathbf{y}_2^k - (1 - \theta_1 - \theta_2)\mathbf{x}_2^k - \theta_2\tilde{\mathbf{x}}_2 - \theta_1\mathbf{x}_2^*\|^2_{\left(\alpha L_2 + \frac{\beta\|\mathbf{A}_2^T\mathbf{A}_2\|}{\theta_1}\right)\mathbf{I}}$$

$$- \frac{1}{2}\mathbb{E}_{i_k}\left(\|\mathbf{x}_2^{k+1} - (1 - \theta_1 - \theta_2)\mathbf{x}_2^k - \theta_2\tilde{\mathbf{x}}_2 - \theta_1\mathbf{x}_2^*\|^2_{\left(\alpha L_2 + \frac{\beta\|\mathbf{A}_2^T\mathbf{A}_2\|}{\theta_1}\right)\mathbf{I}}\right),$$

Step: 5

In step 5, we will first divide $\theta_1$ on both side of Eq. (6) and then summing it with $k$ from 0 to $m-1$. Then after some simplifying, we can obtain

$$\frac{1}{\theta_{1,s}}\mathbb{E}\left(L(\mathbf{x}_s^m, \boldsymbol{\lambda}^*) - L(\mathbf{x}^*, \boldsymbol{\lambda}^*)\right) + \frac{\theta_2 + \theta_{1,s}}{\theta_{1,s}}\sum_{k=1}^{m-1}\mathbb{E}\left(L(\mathbf{x}_s^k, \boldsymbol{\lambda}^*) - L(\mathbf{x}^*, \boldsymbol{\lambda}^*)\right)$$

$$\leq \frac{1}{\theta_{1,s-1}}\mathbb{E}\left(L(\mathbf{x}_{s-1}^m, \boldsymbol{\lambda}^*) - L(\mathbf{x}^*, \boldsymbol{\lambda}^*)\right) + \frac{\theta_2 + \theta_{1,s-1}}{\theta_{1,s-1}}\sum_{k=1}^{m-1}\mathbb{E}\left(L(\mathbf{x}_{s-1}^k, \boldsymbol{\lambda}^*) - L(\mathbf{x}^*, \boldsymbol{\lambda}^*)\right)$$

$$+ \frac{1}{2}\mathbb{E}\|\frac{\mathbf{y}_{s,1}^0 - \theta_2\tilde{\mathbf{x}}_{s,1} - (1 - \theta_{1,s} - \theta_2)\mathbf{x}_{s,1}^0}{\theta_{1,s}} - \mathbf{x}_1^*\|^2_{(\theta_{1,s}L_1 + \|\mathbf{A}_1^T\mathbf{A}_1\|)\mathbf{I} - \mathbf{A}_1^T\mathbf{A}_1}$$

$$- \frac{1}{2}\mathbb{E}\|\frac{\mathbf{x}_{s,1}^m - \theta_2\tilde{\mathbf{x}}_{s,1} - (1 - \theta_{1,s} - \theta_2)\mathbf{x}_{s,1}^{m-1}}{\theta_{1,s}} - \mathbf{x}_1^*\|^2_{(\theta_{1,s}L_1 + \|\mathbf{A}_1^T\mathbf{A}_1\|)\mathbf{I} - \mathbf{A}_1^T\mathbf{A}_1}$$

$$+ \frac{1}{2}\mathbb{E}\|\frac{\mathbf{y}_{s,2}^0 - \theta_2\tilde{\mathbf{x}}_{s,2} - (1 - \theta_{1,s} - \theta_2)\mathbf{x}_{s,2}^0}{\theta_{1,s}} - \mathbf{x}_2^*\|^2_{(\alpha\theta_{1,s}L_2 + \|\mathbf{A}_2^T\mathbf{A}_2\|)\mathbf{I}}$$

$$- \frac{1}{2}\mathbb{E}\|\frac{\mathbf{x}_{s,2}^m - \theta_2\tilde{\mathbf{x}}_{s,2} - (1 - \theta_{1,s} - \theta_2)\mathbf{x}_{s,2}^{m-1}}{\theta_{1,s}} - \mathbf{x}_2^*\|^2_{(\alpha\theta_{1,s}L_2 + \|\mathbf{A}_2^T\mathbf{A}_2\|)\mathbf{I}}$$

$$+ \frac{1}{2\beta}\left(\mathbb{E}\|\hat{\boldsymbol{\lambda}}_s^0 - \boldsymbol{\lambda}^*\|^2 - \mathbb{E}\left[\|\hat{\boldsymbol{\lambda}}_s^m - \boldsymbol{\lambda}^*\|^2\right]\right), \qquad (7)$$

where we use $L(\mathbf{x}_s^k, \boldsymbol{\lambda}^*)$ and $L(\tilde{\mathbf{x}}_s, \boldsymbol{\lambda}^*)$ to denote $L(\mathbf{x}_{s,1}^k, \mathbf{x}_{s,2}^k, \boldsymbol{\lambda}^*)$ and $L(\tilde{\mathbf{x}}_{s,1}, \tilde{\mathbf{x}}_{s,2}, \boldsymbol{\lambda}^*)$, respectively. Note that diving $\theta_1$ (not $\theta_1^2$ ) on both side of Eq. (6) enables us to achieve the non-ergodic $O(1/S)$ result.

Step: 6

Summing Eq. (7) with $s$ from 0 to $S - 1$, and simplifying the result, we obtain

Theorem 1:

$$\frac{1}{2\beta}\mathbb{E}\|\frac{\beta m}{\theta_{1,S}}\left(\mathbf{A}\hat{\mathbf{x}}_S-\mathbf{b}\right)-\frac{\beta(m-1)\theta_2}{\theta_{1,0}}\left(\mathbf{A}\mathbf{x}_0^0-\mathbf{b}\right)+\tilde{\boldsymbol{\lambda}}_0^0-\boldsymbol{\lambda}^*\| \tag{8}$$

$$+\frac{m}{\theta_{1,S}}\mathbb{E}\left(F(\hat{\mathbf{x}}_S)-F(\mathbf{x}^*)+\langle\boldsymbol{\lambda}^*,\mathbf{A}\hat{\mathbf{x}}_S-\mathbf{b}\rangle\right)$$

$$\leq \quad C_3\left(F(\mathbf{x}_0^0)-F(\mathbf{x}^*)+\langle\boldsymbol{\lambda}^*,\mathbf{A}\mathbf{x}_0^0-\mathbf{b}\rangle\right)+\frac{1}{2\beta}\|\tilde{\boldsymbol{\lambda}}_0^0+\frac{\beta(1-\theta_{1,0})}{\theta_{1,0}}(\mathbf{A}\mathbf{x}_0^0-\mathbf{b})-\boldsymbol{\lambda}^*\|^2$$

$$+\frac{1}{2}\|\mathbf{x}_{0,1}^0-\mathbf{x}_1^*\|^2_{(\theta_{1,0}L_1+\|\mathbf{A}_1^T\mathbf{A}_1\|)\mathbf{I}-\mathbf{A}_1^T\mathbf{A}_1}+\frac{1}{2}\|\mathbf{x}_{0,2}^0-\mathbf{x}_2^*\|^2_{\left((1+\frac{1}{b\theta_2})\theta_{1,0}L_2+\|\mathbf{A}_2^T\mathbf{A}_2\|\right)\mathbf{I}},$$

where $C_3=\frac{1-\theta_{1,0}+(m-1)\theta_2}{\theta_{1,0}}$.

Step: 7
We prove Corollary 1:

$$\mathbb{E}|F(\hat{\mathbf{x}}_S)-F(\mathbf{x}^*)| \quad \leq \quad O(\frac{1}{S}),$$

$$\mathbb{E}\|\mathbf{A}\hat{\mathbf{x}}_S-\mathbf{b}\| \quad \leq \quad O(\frac{1}{S}). \tag{9}$$

## 2 Proofs

**Bound Variance.** We bound the variance through [1, 4], namely:

$$\mathbb{E}_{i_k}\left(\|\nabla f_2(\mathbf{y}_2^k)-\tilde{\nabla}f_2(\mathbf{y}_2^k)\|^2\right)$$

$$= \quad \mathbb{E}_{i_k}\left(\|\frac{1}{b}\sum_{i_{k,s}\in\mathcal{I}_{(k,s)}}\left(\nabla f_{2,i_{k,s}}(\mathbf{y}_2^k)-\nabla f_{2,i_{k,s}}(\tilde{\mathbf{x}}_2)+\nabla f_2(\tilde{\mathbf{x}}_2)-\nabla f_2(\mathbf{y}_2^k)\right)\|^2\right)$$

$$\overset{a}{=} \quad \frac{1}{b^2}\mathbb{E}_{i_k}\sum_{i_{k,s}\in\mathcal{I}_{k,s}}\left[\|\left(\nabla f_{2,i_{k,s}}(\mathbf{y}_2^k)-\nabla f_{2,i_{k,s}}(\tilde{\mathbf{x}}_2)\right)-\left(\nabla f_2(\mathbf{y}_2^k)-\nabla f_2(\tilde{\mathbf{x}}_2)\right)\|^2\right]$$

$$\overset{b}{\leq} \quad \frac{1}{b^2}\mathbb{E}_{i_k}\sum_{i_{k,s}\in\mathcal{I}_{k,s}}\left(\|\nabla f_{2,i_{k,s}}(\mathbf{y}_2^k)-\nabla f_{2,i_{k,s}}(\tilde{\mathbf{x}}_2)\|^2\right)$$

$$\leq \quad \frac{2L_2}{b^2}\mathbb{E}_{i_k}\sum_{i_{k,s}\in\mathcal{I}_{k,s}}\left[f_{2,i_{k,s}}(\tilde{\mathbf{x}}_2)-f_{2,i_{k,s}}(\mathbf{y}_2^k)-\langle\nabla f_{2,i_{k,s}}(\mathbf{y}_2^k),\tilde{\mathbf{x}}_2-\mathbf{y}_2^k\rangle\right]$$

$$= \quad \frac{2L_2}{b}\left[f_2(\tilde{\mathbf{x}}_2)-f_2(\mathbf{y}_2^k)-\langle\nabla f_2(\mathbf{y}_2^k),\tilde{\mathbf{x}}_2-\mathbf{y}_2^k\rangle\right], \tag{10}$$

where $\mathbb{E}_{i_k}$ indicates that the expectation is taken over the random choice of $\mathcal{I}_{k,s}$, under the condition that $\mathbf{y}_2^k$, $\tilde{\mathbf{x}}_2$ and $\mathbf{x}_2^k$ are known, in equality $\overset{a}{=}$, we use the fact that each $i_{k,s}$ is independent, and

$$\mathbb{E}_{i_k}\left(\nabla f_{2,i_{k,s}}(\mathbf{y}_2^k)-\nabla f_{2,i_{k,s}}(\tilde{\mathbf{x}}_2)\right)-\left(\nabla f_2(\mathbf{y}_2^k)-\nabla f_2(\tilde{\mathbf{x}}_2)\right)=\mathbf{0};$$

the inequality $\overset{b}{\leq}$ uses the property that $\mathbb{E}\|\xi - \mathbb{E}(\xi)\|^2 = \mathbb{E}\|\xi\|^2 - \|\mathbb{E}\xi\|^2 \leq \mathbb{E}\|\xi\|^2$. The proof is taken from [1, 4].

**Proof of Step 1:**

Set $\bar{\boldsymbol{\lambda}}(\mathbf{x}_1, \mathbf{x}_2) = \frac{\beta}{\theta_1}(\mathbf{A}_1\mathbf{x}_1 + \mathbf{A}_2\mathbf{x}_2 - \mathbf{b}) + \boldsymbol{\lambda}^k$. For the optimal solution of $\mathbf{x}_1$ in Eq. (6) of the paper, we have

$$\left(L_1 + \frac{\beta\|\mathbf{A}_1^T\mathbf{A}_1\|}{\theta_1}\right)(\mathbf{x}_1^{k+1} - \mathbf{y}_1^k) + \nabla f_1(\mathbf{y}_1^k) + \mathbf{A}_1^T\bar{\boldsymbol{\lambda}}(\mathbf{y}_1^k, \mathbf{y}_2^k) \in -\partial h_1(\mathbf{x}_1^{k+1}). \quad (11)$$

Since $f_1$ have Lipschitz continuous gradients, we have

$$
\begin{aligned}
f_1(\mathbf{x}_1^{k+1}) &\leq f_1(\mathbf{y}_1^k) + \langle \nabla f_1(\mathbf{y}_1^k), \mathbf{x}_1^{k+1} - \mathbf{y}_1^k \rangle + \frac{L_1}{2}\|\mathbf{x}_1^{k+1} - \mathbf{y}_1^k\|^2 && (12) \\
&\overset{a}{\leq} f_1(\mathbf{u}_1) + \langle \nabla f_1(\mathbf{y}_1^k), \mathbf{x}_1^{k+1} - \mathbf{u}_1 \rangle + \frac{L_1}{2}\|\mathbf{x}_1^{k+1} - \mathbf{y}_1^k\|^2 \\
&\overset{b}{\leq} f_1(\mathbf{u}_1) - \langle \partial h_1(\mathbf{x}_1^{k+1}), \mathbf{x}_1^{k+1} - \mathbf{u}_1 \rangle - \langle \mathbf{A}_1^T\bar{\boldsymbol{\lambda}}(\mathbf{y}_1^k, \mathbf{y}_2^k), \mathbf{x}_1^{k+1} - \mathbf{u}_1 \rangle \\
&\quad - \left(L_1 + \frac{\beta\|\mathbf{A}_1^T\mathbf{A}_1\|}{\theta_1}\right)\langle \mathbf{x}_1^{k+1} - \mathbf{y}_1^k, \mathbf{x}_1^{k+1} - \mathbf{u}_1 \rangle + \frac{L_1}{2}\|\mathbf{x}_1^{k+1} - \mathbf{y}_1^k\|^2,
\end{aligned}
$$

where $\mathbf{u}_1$ is an arbitrary variable; in the inequality $\overset{a}{\leq}$, we use the property that $f_1(\cdot)$ is convex, and so $f_1(\mathbf{y}_1^k) \leq f_1(\mathbf{u}_1) + \langle \nabla f_1(\mathbf{y}_1^k), \mathbf{y}^k - \mathbf{u} \rangle$ and the inequality $\overset{b}{\leq}$ uses Eq. (11). Then for $h_1(\cdot)$ is convex, and so $h_1(\mathbf{x}_1^{k+1}) \leq h_1(\mathbf{u}_1) + \langle \partial h_1(\mathbf{x}_1^{k+1}), \mathbf{x}^{k+1} - \mathbf{u}_1 \rangle$, we have

$$
\begin{aligned}
F_1(\mathbf{x}_1^{k+1}) &\leq F_1(\mathbf{u}_1) - \langle \mathbf{A}_1^T\bar{\boldsymbol{\lambda}}(\mathbf{y}_1^k, \mathbf{y}_2^k), \mathbf{x}_1^{k+1} - \mathbf{u}_1 \rangle + \frac{L_1}{2}\|\mathbf{x}_1^{k+1} - \mathbf{y}_1^k\|^2 \\
&\quad - \left(L_1 + \frac{\beta\|\mathbf{A}_1^T\mathbf{A}_1\|}{\theta_1}\right)\langle \mathbf{x}_1^{k+1} - \mathbf{y}_1^k, \mathbf{x}_1^{k+1} - \mathbf{u}_1 \rangle. && (13)
\end{aligned}
$$

Setting $\mathbf{u}_1$ be $\mathbf{x}_1^k$, $\tilde{\mathbf{x}}_1$ and $\mathbf{x}_1^*$, respectively, then multiplying the three inequalities by $(1 - \theta_1 - \theta_2)$, $\theta_2$, and $\theta_1$, respectively, and adding them, we have

$$
\begin{aligned}
&F_1(\mathbf{x}_1^{k+1}) && (14) \\
\leq\ & (1 - \theta_1 - \theta_2)F_1(\mathbf{x}_1^k) + \theta_2 F_1(\tilde{\mathbf{x}}_1) + \theta_1 F_1(\mathbf{x}_1^*) + \frac{L_1}{2}\|\mathbf{x}_1^{k+1} - \mathbf{y}_1^k\|^2 \\
& - \langle \mathbf{A}_1^T\bar{\boldsymbol{\lambda}}(\mathbf{y}_1^k, \mathbf{y}_2^k), \mathbf{x}_1^{k+1} - (1 - \theta_1 - \theta_2)\mathbf{x}_1^k - \theta_2\tilde{\mathbf{x}} - \theta_1\mathbf{x}_1^* \rangle \\
& - \left(L_1 + \frac{\beta\|\mathbf{A}_1^T\mathbf{A}_1\|}{\theta_1}\right)\langle \mathbf{x}_1^{k+1} - \mathbf{y}_1^k, \mathbf{x}_1^{k+1} - (1 - \theta_1 - \theta_2)\mathbf{x}_1^k - \theta_2\tilde{\mathbf{x}}_1 - \theta_1\mathbf{x}_1^* \rangle \\
\overset{a}{\leq}\ & (1 - \theta_1 - \theta_2)F_1(\mathbf{x}_1^k) + \theta_2 F_1(\tilde{\mathbf{x}}_1) + \theta_1 F_1(\mathbf{x}_1^*) + \frac{L_1}{2}\|\mathbf{x}_1^{k+1} - \mathbf{y}_1^k\|^2 \\
& - \langle \mathbf{A}_1^T\bar{\boldsymbol{\lambda}}(\mathbf{x}_1^{k+1}, \mathbf{y}_2^k), \mathbf{x}_1^{k+1} - (1 - \theta_1 - \theta_2)\mathbf{x}_1^k - \theta_2\tilde{\mathbf{x}}_1 - \theta_1\mathbf{x}_1^* \rangle \\
& - \langle \mathbf{x}_1^{k+1} - \mathbf{y}_1^k, \mathbf{x}_1^{k+1} - (1 - \theta_1 - \theta_2)\mathbf{x}_1^k - \theta_2\tilde{\mathbf{x}}_1 - \theta_1\mathbf{x}^* \rangle_{\left(L_1 + \frac{\beta\|\mathbf{A}_1^T\mathbf{A}_1\|}{\theta_1}\right)\mathbf{I} - \frac{\beta\mathbf{A}_1^T\mathbf{A}_1}{\theta_1}},
\end{aligned}
$$

where in the equality $\overset{a}{\leq}$, we replace $\mathbf{A}_1^T\bar{\boldsymbol{\lambda}}(\mathbf{y}_1^k,\mathbf{y}_2^k)$ to be $\mathbf{A}_1^T\bar{\boldsymbol{\lambda}}(\mathbf{x}_1^{k+1},\mathbf{y}_2^k)-\frac{\beta\mathbf{A}_1^T\mathbf{A}_1}{\theta_1}(\mathbf{x}_1^{k+1}-\mathbf{y}_1^k)$.

**Proof of step 2:**
For the optimal solution of $\mathbf{x}_2$ in Eq, (7) of the paper, we have

$$\left(\alpha L_2+\frac{\beta\|\mathbf{A}_2^T\mathbf{A}_2\|}{\theta_1}\right)\left(\mathbf{x}_2^{k+1}-\mathbf{y}_2^k\right)+\tilde{\nabla}f_2(\mathbf{y}_2^k)+\mathbf{A}_2^T\bar{\boldsymbol{\lambda}}(\mathbf{x}_1^{k+1},\mathbf{y}_2^k)\in-\partial h_2(\mathbf{x}_2^{k+1}),\,(15)$$

where we set $\alpha=1+\frac{1}{b\theta_2}$. Since $f_2$ have Lipschitz continuous gradients, we have

$$f_2(\mathbf{x}_2^{k+1})\ \leq\ f_2(\mathbf{y}_2^k)+\langle\nabla f_2(\mathbf{y}_2^k),\mathbf{x}_2^{k+1}-\mathbf{y}_2^k\rangle+\frac{L_2}{2}\|\mathbf{x}_2^{k+1}-\mathbf{y}_2^k\|^2.\quad(16)$$

We first consider $\langle\nabla f_2(\mathbf{y}_2^k),\mathbf{x}_2^{k+1}-\mathbf{y}_2^k\rangle$.

$$\begin{aligned}
&\quad\langle\nabla f_2(\mathbf{y}_2^k),\mathbf{x}_2^{k+1}-\mathbf{y}_2^k\rangle\\
&\overset{a}{=}\ \langle\nabla f_2(\mathbf{y}_2^k),\mathbf{u}_2-\mathbf{y}_2^k+\mathbf{x}_2^{k+1}-\mathbf{u}_2\rangle\\
&\overset{b}{=}\ \langle\nabla f_2(\mathbf{y}_2^k),\mathbf{u}_2-\mathbf{y}_2^k\rangle-\theta_3\langle\nabla f_2(\mathbf{y}_2^k),\mathbf{y}_2^k-\tilde{\mathbf{x}}_2^s\rangle+\langle\nabla f_2(\mathbf{y}_2^k),\mathbf{z}^{k+1}-\mathbf{u}_2\rangle\\
&=\ \langle\nabla f_2(\mathbf{y}_2^k),\mathbf{u}_2-\mathbf{y}_2^k\rangle-\theta_3\langle\nabla f_2(\mathbf{y}_2^k),\mathbf{y}_2^k-\tilde{\mathbf{x}}_2^s\rangle\\
&\quad+\langle\tilde{\nabla}f_2(\mathbf{y}_2^k),\mathbf{z}^{k+1}-\mathbf{u}_2\rangle+\langle\nabla f_2(\mathbf{y}_2^k)-\tilde{\nabla}f_2(\mathbf{y}_2^k),\mathbf{z}^{k+1}-\mathbf{u}_2\rangle,\qquad(17)
\end{aligned}$$

where in the equality $\overset{a}{=}$, we introduce an arbitrary variable $\mathbf{u}_2$ (we will set it to be $\mathbf{x}_2^k$, $\tilde{\mathbf{x}}_2$, and $\mathbf{x}_2^*$), and in the equality $\overset{b}{=}$, we set $\mathbf{z}^{k+1}=\mathbf{x}_2^{k+1}+\theta_3(\mathbf{y}_2^k-\tilde{\mathbf{x}}_2)$. For $\langle\tilde{\nabla}f_2(\mathbf{y}_2^k),\mathbf{z}^{k+1}-\mathbf{u}_2\rangle$, we have

$$\begin{aligned}
&\quad\langle\tilde{\nabla}f_2(\mathbf{y}_2^k),\mathbf{z}^{k+1}-\mathbf{u}_2\rangle\qquad\qquad\qquad\qquad\qquad\qquad\qquad\qquad\qquad\quad(18)\\
&\overset{a}{=}\ -\left\langle\partial h_2(\mathbf{x}_2^{k+1})+\mathbf{A}_2^T\bar{\boldsymbol{\lambda}}(\mathbf{x}_1^{k+1},\mathbf{y}_2^k)+\left(\alpha L_2+\frac{\beta\|\mathbf{A}_2^T\mathbf{A}_2\|}{\theta_1}\right)\left(\mathbf{x}_2^{k+1}-\mathbf{y}_2^k\right),\mathbf{z}^{k+1}-\mathbf{u}_2\right\rangle\\
&\overset{b}{=}\ -\langle\partial h_2(\mathbf{x}_2^{k+1}),\mathbf{x}_2^{k+1}+\theta_3(\mathbf{y}_2^k-\tilde{\mathbf{x}}_2)-\mathbf{u}_2\rangle\\
&\quad-\left\langle\mathbf{A}_2^T\bar{\boldsymbol{\lambda}}(\mathbf{x}_1^{k+1},\mathbf{y}_2^k)+\left(\alpha L_2+\frac{\beta\|\mathbf{A}_2^T\mathbf{A}_2\|}{\theta_1}\right)\left(\mathbf{x}_2^{k+1}-\mathbf{y}_2^k\right),\mathbf{z}^{k+1}-\mathbf{u}_2\right\rangle\\
&=\ -\langle\partial h_2(\mathbf{x}_2^{k+1}),\mathbf{x}_2^{k+1}+\theta_3(\mathbf{y}_2^k-\mathbf{x}_2^{k+1}+\mathbf{x}_2^{k+1}-\tilde{\mathbf{x}}_2)-\mathbf{u}_2\rangle\\
&\quad-\left\langle\mathbf{A}_2^T\bar{\boldsymbol{\lambda}}(\mathbf{x}_1^{k+1},\mathbf{y}_2^k)+\left(\alpha L_2+\frac{\beta\|\mathbf{A}_2^T\mathbf{A}_2\|}{\theta_1}\right)\left(\mathbf{x}_2^{k+1}-\mathbf{y}_2^k\right),\mathbf{z}^{k+1}-\mathbf{u}_2\right\rangle\\
&\overset{c}{\leq}\ h_2(\mathbf{u}_2)-h_2(\mathbf{x}_2^{k+1})+\theta_3 h_2(\tilde{\mathbf{x}}_2)-\theta_3 h_2(\mathbf{x}_2^{k+1})-\theta_3\langle\partial h_2(\mathbf{x}_2^{k+1}),\mathbf{y}_2^k-\mathbf{x}_2^{k+1}\rangle\\
&\quad-\left\langle\mathbf{A}_2^T\bar{\boldsymbol{\lambda}}(\mathbf{x}_1^{k+1},\mathbf{y}_2^k)+\left(\alpha L_2+\frac{\beta\|\mathbf{A}_2^T\mathbf{A}_2\|}{\theta_1}\right)\left(\mathbf{x}_2^{k+1}-\mathbf{y}_2^k\right),\mathbf{z}^{k+1}-\mathbf{u}_2\right\rangle\\
&\overset{d}{=}\ h_2(\mathbf{u}_2)-h_2(\mathbf{x}_2^{k+1})+\theta_3 h_2(\tilde{\mathbf{x}}_2)-\theta_3 h_2(\mathbf{x}_2^{k+1})\\
&\quad-\left\langle\mathbf{A}_2^T\bar{\boldsymbol{\lambda}}(\mathbf{x}_1^{k+1},\mathbf{y}_2^k)+\left(\alpha L_2+\frac{\beta\|\mathbf{A}_2^T\mathbf{A}_2\|}{\theta_1}\right)\left(\mathbf{x}_2^{k+1}-\mathbf{y}_2^k\right),\mathbf{z}^{k+1}-\mathbf{u}_2\right\rangle\\
&\quad-\theta_3\left\langle\mathbf{A}_2^T\bar{\boldsymbol{\lambda}}(\mathbf{x}_1^{k+1},\mathbf{y}_2^k)+\left(\alpha L_2+\frac{\beta\|\mathbf{A}_2^T\mathbf{A}_2\|}{\theta_1}\right)\left(\mathbf{x}_2^{k+1}-\mathbf{y}_2^k\right)+\tilde{\nabla}f_2(\mathbf{y}_2^k),\mathbf{x}_2^{k+1}-\mathbf{y}_2^k\right\rangle,
\end{aligned}$$

where in the equalities $\stackrel{a}{=}$ and $\stackrel{d}{=}$, we use Eq. (15); the inequality $\stackrel{b}{=}$ uses $\mathbf{z}^{k+1} = \mathbf{x}_2^{k+1} + \theta_3(\mathbf{y}_2^k - \tilde{\mathbf{x}}_2)$; the inequality $\stackrel{c}{\leq}$ uses the properties that:

$$\langle \partial h_2(\mathbf{x}_2^{k+1}), \mathbf{u}_2 - \mathbf{x}_2^{k+1} \rangle \leq h_2(\mathbf{u}_2) - h_2(\mathbf{x}_2^{k+1}),$$

and

$$\langle \partial h_2(\mathbf{x}_2^{k+1}), \tilde{\mathbf{x}}_2 - \mathbf{x}_2^{k+1} \rangle \leq h_2(\tilde{\mathbf{x}}_2) - h_2(\mathbf{x}_2^{k+1}),$$

since $h_2(\cdot)$ is convex. Rearranging terms on Eq. (18) and using $\tilde{\nabla} f_2(\mathbf{y}_2^k) = \nabla f_2(\mathbf{y}_2^k) + \tilde{\nabla} f_2(\mathbf{y}_2^k) - \nabla f_2(\mathbf{y}_2^k)$, we have

$$
\begin{aligned}
& \langle \tilde{\nabla} f_2(\mathbf{y}_2^k), \mathbf{z}^{k+1} - \mathbf{u}_2 \rangle \qquad\qquad\qquad\qquad\qquad\qquad\qquad (19) \\
= \; & h_2(\mathbf{u}_2) - h_2(\mathbf{x}_2^{k+1}) + \theta_3 h_2(\tilde{\mathbf{x}}_2) - \theta_3 h_2(\mathbf{x}_2^{k+1}) \\
& - \left\langle \mathbf{A}_2^T \bar{\boldsymbol{\lambda}}(\mathbf{x}_1^{k+1}, \mathbf{y}_2^k) + \left( \alpha L_2 + \frac{\beta \|\mathbf{A}_2^T \mathbf{A}_2\|}{\theta_1} \right) (\mathbf{x}_2^{k+1} - \mathbf{y}_2^k), \theta_3(\mathbf{x}_2^{k+1} - \mathbf{y}_2^k) + \mathbf{z}^{k+1} - \mathbf{u}_2 \right\rangle \\
& - \theta_3 \left\langle \nabla f_2(\mathbf{y}_2^k) + \tilde{\nabla} f_2(\mathbf{y}_2^k) - \nabla f_2(\mathbf{y}_2^k), \mathbf{x}_2^{k+1} - \mathbf{y}_2^k \right\rangle.
\end{aligned}
$$

Adding Eq. (17) and Eq. (19), and , we obtain

$$
\begin{aligned}
& (1 + \theta_3) \langle \nabla f_2(\mathbf{y}_2^k), \mathbf{x}_2^{k+1} - \mathbf{y}_2^k \rangle \\
\leq \; & \langle \nabla f_2(\mathbf{y}_2^k), \mathbf{u}_2 - \mathbf{y}_2^k \rangle - \theta_3 \langle \nabla f_2(\mathbf{y}_2^k), \mathbf{y}_2^k - \tilde{\mathbf{x}}_2^s \rangle + h_2(\mathbf{u}_2) - h_2(\mathbf{x}_2^{k+1}) + \theta_3 h_2(\tilde{\mathbf{x}}_2) - \theta_3 h_2(\mathbf{x}_2^{k+1}) \\
& - \left\langle \mathbf{A}_2^T \bar{\boldsymbol{\lambda}}(\mathbf{x}_1^{k+1}, \mathbf{y}_2^k) + \left( \alpha L_2 + \frac{\beta \|\mathbf{A}_2^T \mathbf{A}_2\|}{\theta_1} \right) (\mathbf{x}_2^{k+1} - \mathbf{y}_2^k), \mathbf{z}^{k+1} - \mathbf{u}_2 + \theta_3(\mathbf{x}_2^{k+1} - \mathbf{y}_2^k) \right\rangle \\
& + \langle \nabla f_2(\mathbf{y}_2^k) - \tilde{\nabla} f_2(\mathbf{y}_2^k), \theta_3(\mathbf{x}_2^{k+1} - \mathbf{y}_2^k) + \mathbf{z}^{k+1} - \mathbf{u}_2 \rangle. \qquad\qquad (20)
\end{aligned}
$$

Multiplying Eq. (16) by $(1 + \theta_3)$ and then adding Eq. (20), we can eliminate the term $\langle \nabla f_2(\mathbf{y}_2^k), \mathbf{x}_2^{k+1} - \mathbf{y}_2^k \rangle$ and obtain

$$
\begin{aligned}
& (1 + \theta_3) F_2(\mathbf{x}_2^{k+1}) \\
\leq \; & (1 + \theta_3) f_2(\mathbf{y}_2^k) + \langle \nabla f_2(\mathbf{y}_2^k), \mathbf{u}_2 - \mathbf{y}_2^k \rangle - \theta_3 \langle \nabla f_2(\mathbf{y}_2^k), \mathbf{y}_2^k - \tilde{\mathbf{x}}_2 \rangle + h_2(\mathbf{u}_2) + \theta_3 h_2(\tilde{\mathbf{x}}_2) \\
& - \left\langle \mathbf{A}_2^T \bar{\boldsymbol{\lambda}}(\mathbf{x}_1^{k+1}, \mathbf{y}_2^k) + \left( \alpha L_2 + \frac{\beta \|\mathbf{A}_2^T \mathbf{A}_2\|}{\theta_1} \right) (\mathbf{x}_2^{k+1} - \mathbf{y}_2^k), \mathbf{z}^{k+1} - \mathbf{u}_2 + \theta_3(\mathbf{x}_2^{k+1} - \mathbf{y}_2^k) \right\rangle \\
& + \langle \nabla f_2(\mathbf{y}_2^k) - \tilde{\nabla} f_2(\mathbf{y}_2^k), \theta_3(\mathbf{x}_2^{k+1} - \mathbf{y}_2^k) + \mathbf{z}^{k+1} - \mathbf{u}_2 \rangle + \frac{(1 + \theta_3) L_2}{2} \|\mathbf{x}_2^{k+1} - \mathbf{y}_2^k\|^2 \\
\stackrel{a}{\leq} \; & F_2(\mathbf{u}_2) - \theta_3 \langle \nabla f(\mathbf{y}_2^k), \mathbf{y}_2^k - \tilde{\mathbf{x}}_2 \rangle + \theta_3 f_2(\mathbf{y}_2^k) + \theta_3 h_2(\tilde{\mathbf{x}}_2) \\
& - \left\langle \mathbf{A}_2^T \bar{\boldsymbol{\lambda}}(\mathbf{x}_1^{k+1}, \mathbf{y}_2^k) + \left( \alpha L_2 + \frac{\beta \|\mathbf{A}_2^T \mathbf{A}_2\|}{\theta_1} \right) (\mathbf{x}_2^{k+1} - \mathbf{y}_2^k), \mathbf{z}^{k+1} - \mathbf{u}_2 + \theta_3(\mathbf{x}_2^{k+1} - \mathbf{y}_2^k) \right\rangle \\
& + \langle \nabla f(\mathbf{y}_2^k) - \tilde{\nabla} f_2(\mathbf{y}_2^k), \theta_3(\mathbf{x}_2^{k+1} - \mathbf{y}_2^k) + \mathbf{z}^{k+1} - \mathbf{u}_2 \rangle + \frac{(1 + \theta_3) L_2}{2} \|\mathbf{x}_2^{k+1} - \mathbf{y}_2^k\|^2, \qquad (21)
\end{aligned}
$$

where the inequality $\stackrel{a}{\leq}$ uses the property that $\langle \nabla f_2(\mathbf{y}_2^k), \mathbf{u}_2 - \mathbf{y}_2^k \rangle \leq f_2(\mathbf{u}_2) - f_2(\mathbf{y}_2^k)$.

We now consider the term $\langle \nabla f_2(\mathbf{y}_2^k) - \tilde{\nabla} f_2(\mathbf{y}_2^k), \theta_3(\mathbf{x}_2^{k+1} - \mathbf{y}_2^k) + \mathbf{z}^{k+1} - \mathbf{u}_2 \rangle$. We will set $\mathbf{u}_2$ be $\mathbf{x}_2^k$ and $\mathbf{x}_2^*$, they do not depend on $\mathcal{I}_{k,s}$. So we obtain

$$\mathbb{E}_{i_k} \left( \left\langle \nabla f_2(\mathbf{y}_2^k) - \tilde{\nabla} f_2(\mathbf{y}^k), \theta_3(\mathbf{x}_2^{k+1} - \mathbf{y}_2^k) + \mathbf{z}^{k+1} - \mathbf{u}_2 \right\rangle \right) \tag{22}$$

$$= \mathbb{E}_{i_k} \left( \left\langle \nabla f_2(\mathbf{y}_2^k) - \tilde{\nabla} f_2(\mathbf{y}_2^k), \theta_3 \mathbf{z}^{k+1} + \mathbf{z}^{k+1} \right\rangle \right)$$

$$- \mathbb{E}_{i_k} \left( \left\langle \nabla f_2(\mathbf{y}_2^k) - \tilde{\nabla} f_2(\mathbf{y}_2^k), \theta_3^2(\mathbf{y}_2^k - \tilde{\mathbf{x}}_2) + \theta_3 \mathbf{y}_2^k + \mathbf{u}_2 \right\rangle \right)$$

$$\overset{a}{=} (1+\theta_3) \mathbb{E}_{i_k} (\langle \nabla f_2(\mathbf{y}_2^k) - \tilde{\nabla} f_2(\mathbf{y}_2^k), \mathbf{z}^{k+1} \rangle)$$

$$\overset{b}{=} (1+\theta_3) \mathbb{E}_{i_k} (\langle \nabla f_2(\mathbf{y}_2^k) - \tilde{\nabla} f_2(\mathbf{y}_2^k), \mathbf{x}_2^{k+1} \rangle)$$

$$\overset{c}{=} (1+\theta_3) \mathbb{E}_{i_k} (\langle \nabla f_2(\mathbf{y}_2^k) - \tilde{\nabla} f_2(\mathbf{y}_2^k), \mathbf{x}_2^{k+1} - \mathbf{y}_2^k \rangle)$$

$$\overset{d}{\leq} \mathbb{E}_{i_k} \left( \frac{\theta_3 b}{2L_2} \| \nabla f_2(\mathbf{y}_2^k) - \tilde{\nabla} f_2(\mathbf{y}_2^k) \|^2 \right) + \mathbb{E}_{i_k} \left( \frac{(1+\theta_3)^2 L_2}{2\theta_3 b} \| \mathbf{x}_2^{k+1} - \mathbf{y}_2^k \|^2 \right)$$

$$\overset{e}{\leq} \theta_3 \mathbb{E}_{i_k} \left( f_2(\tilde{\mathbf{x}}_2) - f_2(\mathbf{y}_2^k) - \langle \nabla f_2(\mathbf{y}_2^k), \tilde{\mathbf{x}}_2 - \mathbf{y}_2^k \rangle \right) + \mathbb{E}_{i_k} \left( \frac{(1+\theta_3)^2 L_2}{2\theta_3 b} \| \mathbf{x}_2^{k+1} - \mathbf{y}_2^k \|^2 \right),$$

where $\mathbb{E}_{i_k}$ indicates that the expectation is taken over the random samples in the minibatch $\mathcal{I}_{k,s}$; in the equality $\overset{a}{=}$, we use the fact that

$$\mathbb{E}_{i_k} \left( \nabla f_2(\mathbf{y}_2^k) - \tilde{\nabla} f_2(\mathbf{y}_2^k) \right) = \mathbf{0},$$

and $\mathbf{x}_2^k$, $\mathbf{u}_2$, and $\tilde{\mathbf{x}}_2$ are independent of $i_{k,s}$ (are known), so

$$\mathbb{E}_{i_k} \langle \nabla f_2(\mathbf{y}_2^k) - \tilde{\nabla} f_2(\mathbf{y}_2^k), \mathbf{x}_2^k \rangle = 0,$$
$$\mathbb{E}_{i_k} \langle \nabla f_2(\mathbf{y}_2^k) - \tilde{\nabla} f_2(\mathbf{y}_2^k), \mathbf{y}_2^k \rangle = 0,$$
$$\mathbb{E}_{i_k} \langle \nabla f_2(\mathbf{y}_2^k) - \tilde{\nabla} f_2(\mathbf{y}_2^k), \mathbf{u}_2^k \rangle = 0;$$

the inequalities $\overset{b}{\leq}$ and $\overset{c}{\leq}$ hold similarly; the equality $\overset{d}{\leq}$ uses the Cauchy-Schwarz inequality; $\overset{e}{\leq}$ uses Eq. (10). Taking expectation on Eq. (21) and adding Eq. (22), we obtain

$$(1+\theta_3) \mathbb{E}_{i_k} \left( F_2(\mathbf{x}_2^{k+1}) \right)$$

$$\leq -\mathbb{E}_{i_k} \left\langle \mathbf{A}_2^T \bar{\boldsymbol{\lambda}}(\mathbf{x}_1^{k+1}, \mathbf{y}_2^k) + \left( \alpha L_2 + \frac{\beta \| \mathbf{A}_2^T \mathbf{A}_2 \|}{\theta_1} \right) (\mathbf{x}_2^{k+1} - \mathbf{y}_2^k), \mathbf{z}^{k+1} - \mathbf{u}_2 + \theta_3(\mathbf{x}_2^{k+1} - \mathbf{y}_2^k) \right\rangle$$

$$+ F_2(\mathbf{u}_2) + \theta_3 F(\tilde{\mathbf{x}}_2) + \mathbb{E}_{i_k} \left( \frac{(1+\theta_3)(1 + \frac{1+\theta_3}{b\theta_3}) L_2}{2} \| \mathbf{x}_2^{k+1} - \mathbf{y}_2^k \|^2 \right)$$

$$\overset{a}{=} -\mathbb{E}_{i_k} \left\langle \mathbf{A}_2^T \bar{\boldsymbol{\lambda}}(\mathbf{x}_1^{k+1}, \mathbf{y}_2^k) + \left( \alpha L_2 + \frac{\beta \| \mathbf{A}_2^T \mathbf{A}_2 \|}{\theta_1} \right) (\mathbf{x}_2^{k+1} - \mathbf{y}_2^k), (1+\theta_3)\mathbf{x}_2^{k+1} - \theta_3 \tilde{\mathbf{x}}_2 - \mathbf{u}_2 \right\rangle$$

$$+ F_2(\mathbf{u}_2) + \theta_3 F(\tilde{\mathbf{x}}_2) + \mathbb{E}_{i_k} \left( \frac{(1+\theta_3)(1 + \frac{1}{b\theta_2}) L_2}{2} \| \mathbf{x}_2^{k+1} - \mathbf{y}_2^k \|^2 \right), \tag{23}$$

where in equality $\overset{a}{=}$, we use $\mathbf{z}^{k+1} = \mathbf{x}_2^{k+1} + \theta_3(\mathbf{y}_2^k - \tilde{\mathbf{x}}_2)$ and set $\theta_2 = \frac{\theta_3}{1+\theta_3}$.
Setting $\mathbf{u}_2$ be $\mathbf{x}_2^k$ and $\mathbf{x}_2^*$, respectively, then multiplying the two inequalities by $1 - \theta_1(1+\theta_3)$ and $\theta_1(1+\theta_3)$, and adding them, we obtain

$$
\begin{aligned}
&(1+\theta_3)\mathbb{E}_{i_k}\left(F_2(\mathbf{x}_2^{k+1})\right) \\
\leq\quad & -\mathbb{E}_{i_k}\left\langle \mathbf{A}_2^T\bar{\boldsymbol{\lambda}}(\mathbf{x}_1^{k+1},\mathbf{y}_2^k) + \left(\alpha L_2 + \frac{\beta\|\mathbf{A}_2^T\mathbf{A}_2\|}{\theta_1}\right)(\mathbf{x}_2^{k+1}-\mathbf{y}_2^k), (1+\theta_3)\mathbf{x}_2^{k+1} - \theta_3\tilde{\mathbf{x}}_2 \right\rangle \\
& -\mathbb{E}_{i_k}\left\langle \mathbf{A}_2^T\bar{\boldsymbol{\lambda}}(\mathbf{x}_1^{k+1},\mathbf{y}_2^k) + \left(\alpha L_2 + \frac{\beta\|\mathbf{A}_2^T\mathbf{A}_2\|}{\theta_1}\right)(\mathbf{x}_2^{k+1}-\mathbf{y}_2^k), -\left(1-\theta_1(1+\theta_3)\right)\mathbf{x}_2^k \right\rangle \\
& -\mathbb{E}_{i_k}\left\langle \mathbf{A}_2^T\bar{\boldsymbol{\lambda}}(\mathbf{x}_1^{k+1},\mathbf{y}_2^k) + \left(\alpha L_2 + \frac{\beta\|\mathbf{A}_2^T\mathbf{A}_2\|}{\theta_1}\right)(\mathbf{x}_2^{k+1}-\mathbf{y}_2^k), -\left(\theta_1(1+\theta_3)\right)\mathbf{x}_2^* \right\rangle \\
& + \left(1-\theta_1(1+\theta_3)\right)F_2(\mathbf{x}_2^k) + \left(\theta_1(1+\theta_3)\right)F_2(\mathbf{x}_2^*) + \theta_3 F(\tilde{\mathbf{x}}_2) \\
& + \mathbb{E}_{i_k}\left(\frac{(1+\theta_3)(1+\frac{1}{b\theta_2})L_2}{2}\|\mathbf{x}_2^{k+1}-\mathbf{y}_2^k\|^2\right).
\end{aligned}
\tag{24}
$$

Dividing Eq. (24) by $(1+\theta_3)$, we obtain

$$
\begin{aligned}
&\mathbb{E}_{i_k}F_2(\mathbf{x}_2^{k+1}) \\
\leq\quad & -\mathbb{E}_{i_k}\left\langle \mathbf{A}_2^T\bar{\boldsymbol{\lambda}}(\mathbf{x}_1^{k+1},\mathbf{y}_2^k) + \left(\alpha L_2 + \frac{\beta\|\mathbf{A}_2^T\mathbf{A}_2\|}{\theta_1}\right)(\mathbf{x}_2^{k+1}-\mathbf{y}_2^k), \mathbf{x}_2^{k+1} - \theta_2\tilde{\mathbf{x}}_2 \right\rangle \\
& -\mathbb{E}_{i_k}\left\langle \mathbf{A}_2^T\bar{\boldsymbol{\lambda}}(\mathbf{x}_1^{k+1},\mathbf{y}_2^k) + \left(\alpha L_2 + \frac{\beta\|\mathbf{A}_2^T\mathbf{A}_2\|}{\theta_1}\right)(\mathbf{x}_2^{k+1}-\mathbf{y}_2^k), -(1-\theta_2-\theta_1)\mathbf{x}_2^k - \theta_1\mathbf{x}_2^* \right\rangle \\
& + (1-\theta_2-\theta_1)F_2(\mathbf{x}_2^k) + \theta_1 F_2(\mathbf{x}_2^*) + \theta_2 F_2(\tilde{\mathbf{x}}_2) + \mathbb{E}_{i_k}\left(\frac{(1+\frac{1}{b\theta_2})L_2}{2}\|\mathbf{x}_2^{k+1}-\mathbf{y}_2^k\|^2\right), \tag{25}
\end{aligned}
$$

where we use $\theta_2 = \frac{\theta_3}{1+\theta_3}$ and so $\frac{1-\theta_1(1+\theta_3)}{1+\theta_3} = 1 - \theta_2 - \theta_1$.

**Proof of step 3:**
Through Algorithm 1 in the paper, we have

$$
\boldsymbol{\lambda}^k = \tilde{\boldsymbol{\lambda}}^k + \frac{\beta\theta_2}{\theta_1}\left(\mathbf{A}_1\mathbf{x}_1^k + \mathbf{A}_2\mathbf{x}_2^k - \tilde{\mathbf{b}}\right)
\tag{26}
$$

and

$$
\tilde{\boldsymbol{\lambda}}_s^{k+1} = \boldsymbol{\lambda}_s^k + \beta\left(\mathbf{A}_1\mathbf{x}_{s,1}^{k+1} + \mathbf{A}_2\mathbf{x}_{s,2}^{k+1} - \mathbf{b}\right).
\tag{27}
$$

Setting $\hat{\boldsymbol{\lambda}}^k = \tilde{\boldsymbol{\lambda}}^k + \frac{\beta(1-\theta_1)}{\theta_1}(\mathbf{A}_1\mathbf{x}_1^k + \mathbf{A}_2\mathbf{x}_2^k - \mathbf{b})$, we have

$$\hat{\boldsymbol{\lambda}}^{k+1} \tag{28}$$

$$= \tilde{\boldsymbol{\lambda}}^{k+1} + \beta\left(\frac{1}{\theta_1} - 1\right)(\mathbf{A}_1\mathbf{x}_1^{k+1} + \mathbf{A}_2\mathbf{x}_2^{k+1} - \mathbf{b})$$

$$\stackrel{a}{=} \boldsymbol{\lambda}^k + \frac{\beta}{\theta_1}(\mathbf{A}_1\mathbf{x}_1^{k+1} + \mathbf{A}_2\mathbf{x}_2^{k+1} - \mathbf{b})$$

$$\stackrel{b}{=} \bar{\boldsymbol{\lambda}}(\mathbf{x}_1^{k+1}, \mathbf{x}_2^{k+1})$$

$$\stackrel{c}{=} \tilde{\boldsymbol{\lambda}}^k + \frac{\beta}{\theta_1}\left(\mathbf{A}_1\mathbf{x}_1^{k+1} + \mathbf{A}_2\mathbf{x}_2^{k+1} - \mathbf{b} + \theta_2\left(\mathbf{A}_1(\mathbf{x}_2^k - \tilde{\mathbf{x}}_1) + \mathbf{A}_2(\mathbf{x}_2^k - \tilde{\mathbf{x}}_2)\right)\right),$$

where in equality $\stackrel{a}{=}$, we use Eq. (27); the equality $\stackrel{c}{=}$ is obtained through Eq. (26). Considering into $\hat{\boldsymbol{\lambda}}^k = \tilde{\boldsymbol{\lambda}}^k + \frac{\beta(1-\theta_1)}{\theta_1}(\mathbf{A}_1\mathbf{x}_1^k + \mathbf{A}_2\mathbf{x}_2^k - \mathbf{b})$, we obtain

$$\hat{\boldsymbol{\lambda}}^{k+1} - \hat{\boldsymbol{\lambda}}^k$$

$$= \frac{\beta A_1}{\theta_1}\left(\mathbf{x}_1^{k+1} - (1-\theta_1)\mathbf{x}_1^k - \theta_1\mathbf{x}_1^* + \theta_2(\mathbf{x}_1^k - \tilde{\mathbf{x}}_1)\right)$$

$$+\frac{\beta A_2}{\theta_1}\left(\mathbf{x}_2^{k+1} - (1-\theta_1)\mathbf{x}_2^k - \theta_1\mathbf{x}_2^* + \theta_2(\mathbf{x}_2^k - \tilde{\mathbf{x}}_2)\right), \tag{29}$$

where we use the fact that $\mathbf{A}_1\mathbf{x}_1^* + \mathbf{A}_2\mathbf{x}_2^* = \mathbf{b}$. Now we prove $\hat{\boldsymbol{\lambda}}_{s-1}^m = \hat{\boldsymbol{\lambda}}_s^0$ when $s \geq 1$.

$$\hat{\boldsymbol{\lambda}}_s^0$$

$$= \tilde{\boldsymbol{\lambda}}_s^0 + \frac{\beta(1-\theta_{1,s})}{\theta_{1,s}}\left(\mathbf{A}_1\mathbf{x}_{s,1}^m + \mathbf{A}_2\mathbf{x}_{s,2}^m - \mathbf{b}\right)$$

$$\stackrel{a}{=} \tilde{\boldsymbol{\lambda}}_s^0 + \beta\left(\frac{1}{\theta_{1,s-1}} + \tau - 1\right)\left(\mathbf{A}_1\mathbf{x}_{s,1}^m + \mathbf{A}_2\mathbf{x}_{s,2}^m - \mathbf{b}\right)$$

$$\stackrel{b}{=} \boldsymbol{\lambda}_{s-1}^{m-1} - \beta(\tau - 1)\left(\mathbf{A}_1\mathbf{x}_{s,1}^m + \mathbf{A}_2\mathbf{x}_{s,2}^m - \mathbf{b}\right) + \beta\left(\frac{1}{\theta_{1,s-1}} + \tau - 1\right)\left(\mathbf{A}_1\mathbf{x}_{s,1}^m + \mathbf{A}_2\mathbf{x}_{s,2}^m - \mathbf{b}\right)$$

$$= \boldsymbol{\lambda}_{s-1}^{m-1} + \frac{\beta}{\theta_{1,s-1}}\left(\mathbf{A}_1\mathbf{x}_{s,1}^m + \mathbf{A}_2\mathbf{x}_{s,2}^m - \mathbf{b}\right)$$

$$\stackrel{c}{=} \tilde{\boldsymbol{\lambda}}_{s-1}^m - (\beta - \frac{\beta}{\theta_{1,s-1}})\left(\mathbf{A}_1\mathbf{x}_{s,1}^m + \mathbf{A}_2\mathbf{x}_{s,2}^m - \mathbf{b}\right) = \hat{\boldsymbol{\lambda}}_{s-1}^m, \tag{30}$$

where the equality $\stackrel{a}{=}$ uses the fact that $\frac{1}{\theta_{1,s}} = \frac{1}{\theta_{1,s-1}} + \tau$; the equality $\stackrel{b}{=}$ uses $\tilde{\boldsymbol{\lambda}}_{s+1}^0 = \boldsymbol{\lambda}_s^{m-1} + \beta(1-\tau)(\mathbf{A}_1\mathbf{x}_{s,1}^m + \mathbf{A}_2\mathbf{x}_{s,2}^m - \mathbf{b})$ in Algorithm 2 of the paper; the equality $\stackrel{c}{=}$ uses Eq. (27).

**Proof of Lemma 1:**
Define $L(\mathbf{x}_1, \mathbf{x}_2, \boldsymbol{\lambda}) = F_1(\mathbf{x}_1) - F_1(\mathbf{x}_1^*) + F_2(\mathbf{x}_2) - F_2(\mathbf{x}_2^*) + \langle \boldsymbol{\lambda}, \mathbf{A}_1\mathbf{x}_1 + \mathbf{A}_2\mathbf{x}_2 - \mathbf{b}\rangle$.

We have

$$L(\mathbf{x}_1^{k+1}, \mathbf{x}_2^{k+1}, \boldsymbol{\lambda}^*) - \theta_2 L(\tilde{\mathbf{x}}_1, \tilde{\mathbf{x}}_2, \boldsymbol{\lambda}^*) - (1 - \theta_1 - \theta_2)L(\mathbf{x}_1^k, \mathbf{x}_2^k, \boldsymbol{\lambda}^*)$$
$$= F_1(\mathbf{x}_1^{k+1}) - (1 - \theta_2 - \theta_1)F_1(\mathbf{x}_1^k) - \theta_1 F_1(\mathbf{x}_1^*) - \theta_2 F_1(\tilde{\mathbf{x}}_1)$$
$$+ F_2(\mathbf{x}_2^{k+1}) - (1 - \theta_2 - \theta_1)F_2(\mathbf{x}_2^k) - \theta_1 F_2(\mathbf{x}_2^*) - \theta_2 F_2(\tilde{\mathbf{x}}_2)$$
$$+ \langle \boldsymbol{\lambda}^*, \mathbf{A}_1 \left[ \mathbf{x}_1^{k+1} - (1 - \theta_1 - \theta_2)\mathbf{x}_1^k - \theta_2 \tilde{\mathbf{x}}_1 - \theta_1 \mathbf{x}_1^* \right] \rangle$$
$$+ \langle \boldsymbol{\lambda}^*, \mathbf{A}_2 \left[ \mathbf{x}_2^{k+1} - (1 - \theta_1 - \theta_2)\mathbf{x}_2^k - \theta_2 \tilde{\mathbf{x}}_2 - \theta_1 \mathbf{x}_2^* \right] \rangle. \qquad (31)$$

Adding Eq. (14) and Eq. (25), we have

$$\mathbb{E}_{i_k} \left( L(\mathbf{x}_1^{k+1}, \mathbf{x}_2^{k+1}, \boldsymbol{\lambda}^*) \right) - \theta_2 L(\tilde{\mathbf{x}}_1, \tilde{\mathbf{x}}_2, \boldsymbol{\lambda}^*) - (1 - \theta_2 - \theta_1)L(\mathbf{x}_1^k, \mathbf{x}_2^k, \boldsymbol{\lambda}^*)$$
$$\leq \mathbb{E}_{i_k} \langle \boldsymbol{\lambda}^* - \bar{\boldsymbol{\lambda}}(\mathbf{x}_1^{k+1}, \mathbf{y}_2^k), \mathbf{A}_1 \left[ \mathbf{x}_1^{k+1} - (1 - \theta_1 - \theta_2)\mathbf{x}_1^k - \theta_2 \tilde{\mathbf{x}}_1 - \theta_1 \mathbf{x}_1^* \right] \rangle$$
$$+ \mathbb{E}_{i_k} \langle \boldsymbol{\lambda}^* - \bar{\boldsymbol{\lambda}}(\mathbf{x}_1^{k+1}, \mathbf{y}_2^k), \mathbf{A}_2 \left[ \mathbf{x}_2^{k+1} - (1 - \theta_1 - \theta_2)\mathbf{x}_2^k - \theta_2 \tilde{\mathbf{x}}_2 - \theta_1 \mathbf{x}_2^* \right] \rangle$$
$$- \mathbb{E}_{i_k} \langle \mathbf{x}_1^{k+1} - \mathbf{y}_1^k, \mathbf{x}_1^{k+1} - (1 - \theta_1 - \theta_2)\mathbf{x}_1^k - \theta_2 \tilde{\mathbf{x}}_1 - \theta_1 \mathbf{x}_1^* \rangle_{\left( L_1 + \frac{\beta \|\mathbf{A}_1^T \mathbf{A}_1\|}{\theta_1} \right)\mathbf{I} - \frac{\beta \mathbf{A}_1^T \mathbf{A}_1}{\theta_1}}$$
$$- \mathbb{E}_{i_k} \langle \mathbf{x}_2^{k+1} - \mathbf{y}_2^k, \mathbf{x}_2^{k+1} - \theta_2 \tilde{\mathbf{x}}_2 - (1 - \theta_2 - \theta_1)\mathbf{x}_2^k - \theta_1 \mathbf{x}^* \rangle_{\left( \alpha L_2 + \frac{\beta \|\mathbf{A}_2^T \mathbf{A}_2\|}{\theta_1} \right)\mathbf{I}}$$
$$+ \frac{L_1}{2} \mathbb{E}_{i_k} \|\mathbf{x}_1^{k+1} - \mathbf{y}_1^k\|^2 + \mathbb{E}_{i_k} \left( \frac{(1 + \frac{1}{b\theta_2})L_2}{2} \|\mathbf{x}_2^{k+1} - \mathbf{y}_2^k\|^2 \right)$$
$$\overset{a}{=} \mathbb{E}_{i_k} \langle \boldsymbol{\lambda}^* - \bar{\boldsymbol{\lambda}}(\mathbf{x}_1^{k+1}, \mathbf{x}_2^{k+1}), \mathbf{A}_1 \left[ \mathbf{x}_1^{k+1} - (1 - \theta_1 - \theta_2)\mathbf{x}_1^k - \theta_2 \tilde{\mathbf{x}}_1 - \theta_1 \mathbf{x}_1^* \right] \rangle$$
$$+ \mathbb{E}_{i_k} \langle \boldsymbol{\lambda}^* - \bar{\boldsymbol{\lambda}}(\mathbf{x}_1^{k+1}, \mathbf{x}_2^{k+1}), \mathbf{A}_2 \left[ \mathbf{x}_2^{k+1} - (1 - \theta_1 - \theta_3)\mathbf{x}_2^k - \theta_2 \tilde{\mathbf{x}}_2 - \theta_1 \mathbf{x}_2^* \right] \rangle$$
$$- \mathbb{E}_{i_k} \langle \mathbf{x}_1^{k+1} - \mathbf{y}_1^k, \mathbf{x}_1^{k+1} - (1 - \theta_1 - \theta_2)\mathbf{x}_1^k - \theta_2 \tilde{\mathbf{x}}_1 - \theta_1 \mathbf{x}_1^* \rangle_{\left( L_1 + \frac{\beta \|\mathbf{A}_1^T \mathbf{A}_1\|}{\theta_1} \right)\mathbf{I} - \frac{\beta \mathbf{A}_1^T \mathbf{A}_1}{\theta_1}}$$
$$- \mathbb{E}_{i_k} \langle \mathbf{x}_2^{k+1} - \mathbf{y}_2^k, \mathbf{x}_2^{k+1} - \theta_2 \tilde{\mathbf{x}}_2 - (1 - \theta_2 - \theta_1)\mathbf{x}_2^k - \theta_1 \mathbf{x}^* \rangle_{\left( \alpha L_2 + \frac{\beta \|\mathbf{A}_2^T \mathbf{A}_2\|}{\theta_1} \right)\mathbf{I} - \frac{\beta \mathbf{A}_2^T \mathbf{A}_2}{\theta_1}}$$
$$+ \frac{L_1}{2} \mathbb{E}_{i_k} \|\mathbf{x}_1^{k+1} - \mathbf{y}_1^k\|^2 + \mathbb{E}_{i_k} \left( \frac{(1 + \frac{1}{b\theta_2})L_2}{2} \|\mathbf{x}_2^{k+1} - \mathbf{y}_2^k\|^2 \right)$$
$$+ \frac{\beta}{\theta_1} \mathbb{E}_{i_k} \langle \mathbf{A}_2 \mathbf{x}_2^{k+1} - \mathbf{A}_2 \mathbf{y}_2^k, \mathbf{A}_1 \left[ \mathbf{x}_1^{k+1} - (1 - \theta_1 - \theta_2)\mathbf{x}_1^k - \theta_2 \tilde{\mathbf{x}}_1 - \theta_1 \mathbf{x}_1^* \right] \rangle, \qquad (32)$$

where in the equality $\overset{a}{=}$, we change the term $\bar{\boldsymbol{\lambda}}(\mathbf{x}_1^{k+1}, \mathbf{y}_2^k)$ to $\bar{\boldsymbol{\lambda}}(\mathbf{x}_1^{k+1}, \mathbf{x}_2^{k+1}) - \frac{\beta \mathbf{A}_2^T \mathbf{A}_2}{\theta_1}(\mathbf{x}_2^{k+1} - \mathbf{y}_2^k)$. For the first two terms in the right hand of Eq. (32), we have

$$\langle \boldsymbol{\lambda}^* - \bar{\boldsymbol{\lambda}}(\mathbf{x}_1^{k+1}, \mathbf{x}_2^{k+1}), \mathbf{A}_1 \left[ \mathbf{x}_1^{k+1} - (1 - \theta_1 - \theta_2)\mathbf{x}_1^k - \theta_2 \tilde{\mathbf{x}}_1 - \theta_1 \mathbf{x}_1^* \right] \rangle$$
$$+ \langle \boldsymbol{\lambda}^* - \bar{\boldsymbol{\lambda}}(\mathbf{x}_1^{k+1}, \mathbf{x}_2^{k+1}), \mathbf{A}_2 \left[ \mathbf{x}_2^{k+1} - (1 - \theta_1 - \theta_2)\mathbf{x}_1^k - \theta_2 \tilde{\mathbf{x}}_2 - \theta_1 \mathbf{x}_2^* \right] \rangle$$
$$= \frac{\theta_1}{\beta} \langle \boldsymbol{\lambda}^* - \hat{\boldsymbol{\lambda}}^{k+1}, \hat{\boldsymbol{\lambda}}^{k+1} - \hat{\boldsymbol{\lambda}}^k \rangle$$
$$= \frac{\theta_1}{2\beta} \left( \|\hat{\boldsymbol{\lambda}}^k - \boldsymbol{\lambda}^*\|^2 - \|\hat{\boldsymbol{\lambda}}^{k+1} - \boldsymbol{\lambda}^*\|^2 - \|\hat{\boldsymbol{\lambda}}^{k+1} - \hat{\boldsymbol{\lambda}}^k\|^2 \right). \qquad (33)$$

where in the first equality, we use $\overset{b}{=}$ in Eq. (28) and Eq. (29), and in the second equality we use the fact that

$$\langle \mathbf{a} - \mathbf{b}, \mathbf{a} - \mathbf{c} \rangle = \frac{1}{2}\|\mathbf{a} - \mathbf{b}\|^2 + \frac{1}{2}\|\mathbf{a} - \mathbf{c}\|^2 - \frac{1}{2}\|\mathbf{b} - \mathbf{c}\|^2.$$

Substituting Eq (33) into Eq. (32), we obtain:

$$
\begin{aligned}
&\mathbb{E}_{i_k}\left(L(\mathbf{x}_1^{k+1}, \mathbf{x}_2^{k+1}, \boldsymbol{\lambda}^*)\right) - \theta_2 L(\tilde{\mathbf{x}}_1, \tilde{\mathbf{x}}_2, \boldsymbol{\lambda}^*) - (1 - \theta_2 - \theta_1)L(\mathbf{x}_1^k, \mathbf{x}_2^k, \boldsymbol{\lambda}^*) \\
\leq\ & \frac{\theta_1}{2\beta}\left(\|\hat{\boldsymbol{\lambda}}^k - \boldsymbol{\lambda}^*\|^2 - \mathbb{E}_{i_k}\|\hat{\boldsymbol{\lambda}}^{k+1} - \boldsymbol{\lambda}^*\|^2 - \mathbb{E}_{i_k}\|\hat{\boldsymbol{\lambda}}^{k+1} - \hat{\boldsymbol{\lambda}}^k\|^2\right) \\
&+ \mathbb{E}_{i_k}\left\langle \mathbf{x}_1^{k+1} - \mathbf{y}_1^k, \mathbf{x}_1^{k+1} - (1 - \theta_1 - \theta_2)\mathbf{x}_1^k - \theta_2\tilde{\mathbf{x}}_1 - \theta_1\mathbf{x}_1^* \right\rangle_{\left(L_1 + \frac{\beta\|\mathbf{A}_1^T\mathbf{A}_1\|}{\theta_1}\right)\mathbf{I} - \frac{\beta\mathbf{A}_1^T\mathbf{A}_1}{\theta_1}} \\
&- \mathbb{E}_{i_k}\left\langle \mathbf{x}_2^{k+1} - \mathbf{y}_2^k, \mathbf{x}_2^{k+1} - \theta_2\tilde{\mathbf{x}}_2 - (1 - \theta_2 - \theta_1)\mathbf{x}_2^k - \theta_1\mathbf{x}^* \right\rangle_{\left(\alpha L_2 + \frac{\beta\|\mathbf{A}_2^T\mathbf{A}_2\|}{\theta_1}\right)\mathbf{I} - \frac{\beta\mathbf{A}_2^T\mathbf{A}_2}{\theta_1}} \\
&+ \frac{L_1}{2}\mathbb{E}_{i_k}\|\mathbf{x}_1^{k+1} - \mathbf{y}_1^k\|^2 + \mathbb{E}_{i_k}\left(\frac{(1 + \frac{1}{b\theta_2})L_2}{2}\|\mathbf{x}_2^{k+1} - \mathbf{y}_2^k\|^2\right) \\
&+ \frac{\beta}{\theta_1}\mathbb{E}_{i_k}\left\langle \mathbf{A}_2\mathbf{x}_2^{k+1} - \mathbf{A}_2\mathbf{y}_2^k, \mathbf{A}_1\left[\mathbf{x}_1^{k+1} - (1 - \theta_1 - \theta_2)\mathbf{x}_1^k - \theta_2\tilde{\mathbf{x}}_1 - \theta_1\mathbf{x}_1^*\right]\right\rangle. \qquad (34)
\end{aligned}
$$

For the fourth and fifth terms in the right hand of Eq. (34), we have

$$
\begin{aligned}
&\langle \mathbf{x}_i^{k+1} - \mathbf{y}_i^k, \mathbf{x}_i^{k+1} - (1 - \theta_1 - \theta_2)\mathbf{x}_i^k - \theta_2\tilde{\mathbf{x}}_i - \theta_1\mathbf{x}_i^* \rangle_{\mathbf{G}_i} \\
\leq\ & \frac{1}{2}\left(\|\mathbf{x}_i^{k+1} - (1 - \theta_1 - \theta_2)\mathbf{x}_i^k - \theta_2\tilde{\mathbf{x}}_i - \theta_1\mathbf{x}_i^*\|_{\mathbf{G}_i}^2 + \|\mathbf{x}_i^{k+1} - \mathbf{y}_i^k\|_{\mathbf{G}_i}^2\right) \\
&- \frac{1}{2}\|\mathbf{y}_i^k - (1 - \theta_1 - \theta_2)\mathbf{x}_i^k - \theta_2\tilde{\mathbf{x}}_i - \theta_1\mathbf{x}_i^*\|_{\mathbf{G}_i}^2, \quad i = 1, 2, \qquad (35)
\end{aligned}
$$

where $\mathbf{G}_1 = \left(L_1 + \frac{\beta\|\mathbf{A}_1^T\mathbf{A}_1\|}{\theta_1}\right)\mathbf{I} - \frac{\beta\mathbf{A}_1^T\mathbf{A}_1}{\theta_1}$ and $\mathbf{G}_2 = \left(\alpha L_2 + \frac{\beta\|\mathbf{A}_2^T\mathbf{A}_2\|}{\theta_1}\right)\mathbf{I} - $

$\frac{\beta \mathbf{A}_2^T \mathbf{A}_2}{\theta_1}$. Then substituting Eq (35) into Eq. (34), we obtain:

$$\mathbb{E}_{i_k}\left(L(\mathbf{x}_1^{k+1}, \mathbf{x}_2^{k+1}, \boldsymbol{\lambda}^*)\right) - \theta_2 L(\tilde{\mathbf{x}}_1, \tilde{\mathbf{x}}_2, \boldsymbol{\lambda}^*) - (1 - \theta_2 - \theta_1)L(\mathbf{x}_1^k, \mathbf{x}_2^k, \boldsymbol{\lambda}^*) \qquad (36)$$

$$\leq \quad \frac{\theta_1}{2\beta}\left(\|\hat{\boldsymbol{\lambda}}^k - \boldsymbol{\lambda}^*\|^2 - \mathbb{E}_{i_k}\|\hat{\boldsymbol{\lambda}}^{k+1} - \boldsymbol{\lambda}^*\|^2 - \mathbb{E}_{i_k}\|\hat{\boldsymbol{\lambda}}^{k+1} - \hat{\boldsymbol{\lambda}}^k\|^2\right)$$

$$+ \frac{1}{2}\|\mathbf{y}_1^k - (1 - \theta_1 - \theta_2)\mathbf{x}_1^k - \theta_2\tilde{\mathbf{x}}_1 - \theta_1\mathbf{x}_1^*\|^2_{\left(L_1 + \frac{\|\beta \mathbf{A}_1^T \mathbf{A}_1\|}{\theta_1}\right)\mathbf{I} - \frac{\beta \mathbf{A}_1^T \mathbf{A}_1}{\theta_1}}$$

$$- \frac{1}{2}\mathbb{E}_{i_k}\left(\|\mathbf{x}_1^{k+1} - (1 - \theta_1 - \theta_2)\mathbf{x}_1^k - \theta_2\tilde{\mathbf{x}}_1 - \theta_1\mathbf{x}_1^*\|^2_{\left(L_1 + \frac{\beta \|\mathbf{A}_1^T \mathbf{A}_1\|}{\theta_1}\right)\mathbf{I} - \frac{\beta \mathbf{A}_1^T \mathbf{A}_1}{\theta_1}}\right)$$

$$+ \frac{1}{2}\|\mathbf{y}_2^k - (1 - \theta_1 - \theta_2)\mathbf{x}_2^k - \theta_2\tilde{\mathbf{x}}_2 - \theta_1\mathbf{x}_2^*\|^2_{\left(\alpha L_2 + \frac{\beta \|\mathbf{A}_2^T \mathbf{A}_2\|}{\theta_1}\right)\mathbf{I} - \frac{\beta \mathbf{A}_2^T \mathbf{A}_2}{\theta_1}}$$

$$- \frac{1}{2}\mathbb{E}_{i_k}\left(\|\mathbf{x}_2^{k+1} - (1 - \theta_1 - \theta_2)\mathbf{x}_2^k - \theta_2\tilde{\mathbf{x}}_2 - \theta_1\mathbf{x}_2^*\|^2_{\left(\alpha L_2 + \frac{\beta \|\mathbf{A}_2^T \mathbf{A}_2\|}{\theta_1}\right)\mathbf{I} - \frac{\beta \mathbf{A}_2^T \mathbf{A}_2}{\theta_1}}\right)$$

$$- \mathbb{E}_{i_k}\|\mathbf{x}_1^{k+1} - \mathbf{y}_1^k\|^2_{\left(\frac{\beta \|\mathbf{A}_1^T \mathbf{A}_1\|}{\theta_1}\right)\mathbf{I} - \frac{\beta \mathbf{A}_1^T \mathbf{A}_1}{\theta_1}} - \mathbb{E}_{i_k}\|\mathbf{x}_2^{k+1} - \mathbf{y}_2^k\|^2_{\left(\frac{\beta \|\mathbf{A}_2^T \mathbf{A}_2\|}{\theta_1}\right)\mathbf{I} - \frac{\beta \mathbf{A}_2^T \mathbf{A}_2}{\theta_1}}$$

$$+ \frac{\beta}{\theta_1}\mathbb{E}_{i_k}\left\langle \mathbf{A}_2\mathbf{x}_2^{k+1} - \mathbf{A}_2\mathbf{y}_2^k, \mathbf{A}_1\left[\mathbf{x}_1^{k+1} - (1 - \theta_1 - \theta_2)\mathbf{x}_1^k - \theta_2\tilde{\mathbf{x}}_1 - \theta_1\mathbf{x}_1^*\right]\right\rangle.$$

For the last term in the right hand of Eq. (36), we have

$$\frac{\beta}{\theta_1}\left\langle \mathbf{A}_2\mathbf{x}_2^{k+1} - \mathbf{A}_2\mathbf{y}_2^k, \mathbf{A}_1\left[\mathbf{x}_1^{k+1} - (1 - \theta_1 - \theta_2)\mathbf{x}_1^k - \theta_2\tilde{\mathbf{x}}_1 - \theta_1\mathbf{x}_1^*\right]\right\rangle$$

$$\stackrel{a}{=} \frac{\beta}{\theta_1}\left\langle \mathbf{A}_2\mathbf{x}_2^{k+1} - \mathbf{A}_2\mathbf{v} - (\mathbf{A}_2\mathbf{y}_2^k - \mathbf{A}_2\mathbf{v}), \mathbf{A}_1\left[\mathbf{x}_1^{k+1} - (1 - \theta_1 - \theta_2)\mathbf{x}_1^k - \theta_2\tilde{\mathbf{x}}_1 - \theta_1\mathbf{x}_1^*\right] - \mathbf{0}\right\rangle$$

$$\stackrel{b}{=} \frac{\beta}{2\theta_1}\|\mathbf{A}_2\mathbf{x}_2^{k+1} - \mathbf{A}_2\mathbf{v} + \mathbf{A}_1\left[\mathbf{x}_1^{k+1} - (1 - \theta_1 - \theta_2)\mathbf{x}_1^k - \theta_2\tilde{\mathbf{x}}_1 - \theta_1\mathbf{x}_1^*\right]\|^2$$

$$- \frac{\beta}{2\theta_1}\|\mathbf{A}_2\mathbf{x}_2^{k+1} - \mathbf{A}_2\mathbf{v}\|^2 + \frac{\beta}{2\theta_1}\|\mathbf{A}_2\mathbf{y}_2^k - \mathbf{A}_2\mathbf{v}\|^2$$

$$- \frac{\beta}{2\theta_1}\|\mathbf{A}_2\mathbf{y}_2^k - \mathbf{A}_2\mathbf{v} + \mathbf{A}_1\left(\mathbf{x}_1^{k+1} - (1 - \theta_1 - \theta_2)\mathbf{x}_1^k - \theta_2\tilde{\mathbf{x}}_1 - \theta_1\mathbf{x}_1^*\right)\|^2,$$

$$\stackrel{c}{=} \frac{\theta_1}{2\beta}\|\hat{\boldsymbol{\lambda}}^{k+1} - \hat{\boldsymbol{\lambda}}^k\|^2 - \frac{\beta}{2\theta_1}\|\mathbf{A}_2\mathbf{x}_2^{k+1} - \mathbf{A}_2\mathbf{v}\|^2 + \frac{\beta}{2\theta_1}\|\mathbf{A}_2\mathbf{y}_2^k - \mathbf{A}_2\mathbf{v}\|^2$$

$$- \frac{\beta}{2\theta_1}\|\mathbf{A}_2\mathbf{y}_2^k - \mathbf{A}_2\mathbf{v} + \mathbf{A}_1\left(\mathbf{x}_1^{k+1} - (1 - \theta_1 - \theta_2)\mathbf{x}_1^k - \theta_2\tilde{\mathbf{x}}_1 - \theta_1\mathbf{x}_1^*\right)\|^2, \qquad (37)$$

where in the equality $\stackrel{a}{=}$, we set $\mathbf{v} = (1 - \theta_1 - \theta_2)\mathbf{x}_2^k + \theta_2\tilde{\mathbf{x}}_2 + \theta_1\mathbf{x}_2^*$; the equality $\stackrel{b}{=}$ uses the fact that

$$\langle \mathbf{a} - \mathbf{b}, \mathbf{c} - \mathbf{d}\rangle = \frac{1}{2}\left(\|\mathbf{a} + \mathbf{c}\|^2 - \|\mathbf{a} + \mathbf{d}\|^2 + \|\mathbf{b} + \mathbf{d}\|^2 - \|\mathbf{b} + \mathbf{c}\|^2\right),$$

and the equality $\overset{c}{=}$ uses Eq. (28). Substituting Eq. (37) into Eq. (36), we have

$$\mathbb{E}_{i_k}\left(L(\mathbf{x}_1^{k+1},\mathbf{x}_2^{k+1},\boldsymbol{\lambda}^*)\right) - \theta_2 L(\tilde{\mathbf{x}}_1,\tilde{\mathbf{x}}_2,\boldsymbol{\lambda}^*) - (1-\theta_2-\theta_1)L(\mathbf{x}_1^k,\mathbf{x}_2^k,\boldsymbol{\lambda}^*) \quad (38)$$

$$\leq \frac{\theta_1}{2\beta}\left(\|\hat{\boldsymbol{\lambda}}^k - \boldsymbol{\lambda}^*\|^2 - \mathbb{E}_{i_k}\|\hat{\boldsymbol{\lambda}}^{k+1} - \boldsymbol{\lambda}^*\|^2\right)$$

$$+\frac{1}{2}\|\mathbf{y}_1^k - (1-\theta_1-\theta_2)\mathbf{x}_1^k - \theta_2\tilde{\mathbf{x}}_1 - \theta_1\mathbf{x}_1^*\|^2_{\left(L_1+\frac{\beta\|\mathbf{A}_1^T\mathbf{A}_1\|}{\theta_1}\right)\mathbf{I}-\frac{\beta\mathbf{A}_1^T\mathbf{A}_1}{\theta_1}}$$

$$-\frac{1}{2}\mathbb{E}_{i_k}\left(\|\mathbf{x}_1^{k+1} - (1-\theta_1-\theta_2)\mathbf{x}_1^k - \theta_2\tilde{\mathbf{x}}_1 - \theta_1\mathbf{x}_1^*\|^2_{\left(L_1+\frac{\beta\|\mathbf{A}_1^T\mathbf{A}_1\|}{\theta_1}\right)\mathbf{I}-\frac{\beta\mathbf{A}_1^T\mathbf{A}_1}{\theta_1}}\right)$$

$$+\frac{1}{2}\|\mathbf{y}_2^k - (1-\theta_1-\theta_2)\mathbf{x}_2^k - \theta_2\tilde{\mathbf{x}}_2 - \theta_1\mathbf{x}_2^*\|^2_{\left(\alpha L_2+\frac{\beta\|\mathbf{A}_2^T\mathbf{A}_2\|}{\theta_1}\right)\mathbf{I}}$$

$$-\mathbb{E}_{i_k}\|\mathbf{x}_1^{k+1} - \mathbf{y}_1^k\|^2_{\left(\frac{\beta\|\mathbf{A}_1^T\mathbf{A}_1\|}{\theta_1}\right)\mathbf{I}-\frac{\beta\mathbf{A}_1^T\mathbf{A}_1}{\theta_1}} - \mathbb{E}_{i_k}\|\mathbf{x}_2^{k+1} - \mathbf{y}_2^k\|^2_{\left(\frac{\beta\|\mathbf{A}_2^T\mathbf{A}_2\|}{\theta_1}\right)\mathbf{I}-\frac{\beta\mathbf{A}_2^T\mathbf{A}_2}{\theta_1}}$$

$$-\frac{\beta}{2\theta_1}\mathbb{E}_{i_k}\|\mathbf{A}_2\mathbf{y}_2^k - \mathbf{A}_2\mathbf{v} + \mathbf{A}_1\left(\mathbf{x}_1^{k+1} - (1-\theta_1-\theta_2)\mathbf{x}_1^k - \theta_2\tilde{\mathbf{x}}_1 - \theta_1\mathbf{x}_1^*\right)\|^2.$$

Since the last three terms in the right hand of Eq. (38) are nonpositive, we obtain:

$$\mathbb{E}_{i_k}\left(L(\mathbf{x}_1^{k+1},\mathbf{x}_2^{k+1},\boldsymbol{\lambda}^*)\right) - \theta_2 L(\tilde{\mathbf{x}}_1,\tilde{\mathbf{x}}_2,\boldsymbol{\lambda}^*) - (1-\theta_2-\theta_1)L(\mathbf{x}_1^k,\mathbf{x}_2^k,\boldsymbol{\lambda}^*) \quad (39)$$

$$\leq \frac{\theta_1}{2\beta}\left(\|\hat{\boldsymbol{\lambda}}^k - \boldsymbol{\lambda}^*\|^2 - \mathbb{E}_{i_k}\|\hat{\boldsymbol{\lambda}}^{k+1} - \boldsymbol{\lambda}^*\|^2\right)$$

$$+\frac{1}{2}\|\mathbf{y}_1^k - (1-\theta_1-\theta_2)\mathbf{x}_1^k - \theta_2\tilde{\mathbf{x}}_1 - \theta_1\mathbf{x}_1^*\|^2_{\left(L_1+\frac{\beta\|\mathbf{A}_1^T\mathbf{A}_1\|}{\theta_1}\right)\mathbf{I}-\frac{\beta\mathbf{A}_1^T\mathbf{A}_1}{\theta_1}}$$

$$-\frac{1}{2}\mathbb{E}_{i_k}\left(\|\mathbf{x}_1^{k+1} - (1-\theta_1-\theta_2)\mathbf{x}_1^k - \theta_2\tilde{\mathbf{x}}_1 - \theta_1\mathbf{x}_1^*\|^2_{\left(L_1+\frac{\beta\|\mathbf{A}_1^T\mathbf{A}_1\|}{\theta_1}\right)\mathbf{I}-\frac{\beta\mathbf{A}_1^T\mathbf{A}_1}{\theta_1}}\right)$$

$$+\frac{1}{2}\|\mathbf{y}_2^k - (1-\theta_1-\theta_2)\mathbf{x}_2^k - \theta_2\tilde{\mathbf{x}}_2 - \theta_1\mathbf{x}_2^*\|^2_{\left(\alpha L_2+\frac{\beta\|\mathbf{A}_2^T\mathbf{A}_2\|}{\theta_1}\right)\mathbf{I}}$$

$$-\frac{1}{2}\mathbb{E}_{i_k}\left(\|\mathbf{x}_2^{k+1} - (1-\theta_1-\theta_2)\mathbf{x}_2^k - \theta_2\tilde{\mathbf{x}}_2 - \theta_1\mathbf{x}_2^*\|^2_{\left(\alpha L_2+\frac{\beta\|\mathbf{A}_2^T\mathbf{A}_2\|}{\theta_1}\right)\mathbf{I}}\right).$$

So Lemma 1 is proved.

**Proof of Step 5:**

Taking expectation over the first $k$ iterations for Eq. (38) and diving $\theta_1$ on

sides of it, we obtain:

$$\frac{1}{\theta_1}\mathbb{E}\left[L(\mathbf{x}_1^{k+1},\mathbf{x}_2^{k+1},\boldsymbol{\lambda}^*)\right]-\frac{\theta_2}{\theta_1}L(\tilde{\mathbf{x}}_1,\tilde{\mathbf{x}}_2,\boldsymbol{\lambda}^*)-\frac{1-\theta_2-\theta_1}{\theta_1}L(\mathbf{x}_1^k,\mathbf{x}_2^k,\boldsymbol{\lambda}^*) \quad (40)$$

$$\begin{aligned}
\leq \quad & \frac{1}{2\beta}\left(\|\hat{\boldsymbol{\lambda}}^k-\boldsymbol{\lambda}^*\|^2-\mathbb{E}\left[\|\hat{\boldsymbol{\lambda}}^{k+1}-\boldsymbol{\lambda}^*\|^2\right]\right) \\
& +\frac{\theta_1}{2}\|\frac{\mathbf{y}_1^k-(1-\theta_1-\theta_2)\mathbf{x}_1^k-\theta_2\tilde{\mathbf{x}}_1}{\theta_1}-\mathbf{x}_1^*\|^2_{\left(L_1+\frac{\|\mathbf{A}_1^T\mathbf{A}_1\|}{\theta_1}\right)\mathbf{I}-\frac{\mathbf{A}_1^T\mathbf{A}_1}{\theta_1}} \\
& -\frac{\theta_1}{2}\mathbb{E}\left(\|\frac{\mathbf{x}_1^{k+1}-(1-\theta_1-\theta_2)\mathbf{x}_1^k-\theta_2\tilde{\mathbf{x}}_1}{\theta_1}-\mathbf{x}_1^*\|^2_{\left(L_1+\frac{\|\mathbf{A}_1^T\mathbf{A}_1\|}{\theta_1}\right)\mathbf{I}-\frac{\mathbf{A}_1^T\mathbf{A}_1}{\theta_1}}\right) \\
& +\frac{\theta_1}{2}\|\frac{\mathbf{y}_2^k-(1-\theta_1-\theta_2)\mathbf{x}_2^k-\theta_2\tilde{\mathbf{x}}_2}{\theta_1}-\mathbf{x}_2^*\|^2_{\left(\alpha L_2+\frac{\|\mathbf{A}_2^T\mathbf{A}_2\|}{\theta_1}\right)\mathbf{I}} \\
& -\frac{\theta_1}{2}\mathbb{E}\left(\|\frac{\mathbf{x}_2^{k+1}-(1-\theta_1-\theta_2)\mathbf{x}_2^k-\theta_2\tilde{\mathbf{x}}_2}{\theta_1}-\mathbf{x}_2^*\|^2_{\left(\alpha L_2+\frac{\|\mathbf{A}_2^T\mathbf{A}_2\|}{\theta_1}\right)\mathbf{I}}\right),
\end{aligned}$$

the expectation is taken under the condition that randomness in the first $s$ epochs are fixed. Since

$$\mathbf{y}^k=\mathbf{x}^k+(1-\theta_1-\theta_2)(\mathbf{x}^k-\mathbf{x}^{k-1}),\quad k\geq 1,$$

we obtain:

$$\frac{1}{\theta_1}\mathbb{E}\left[L(\mathbf{x}_1^{k+1},\mathbf{x}_2^{k+1},\boldsymbol{\lambda}^*)\right]-\frac{\theta_2}{\theta_1}L(\tilde{\mathbf{x}}_1,\tilde{\mathbf{x}}_2,\boldsymbol{\lambda}^*)-\frac{1-\theta_2-\theta_1}{\theta_1}L(\mathbf{x}_1^k,\mathbf{x}_2^k,\boldsymbol{\lambda}^*) \quad (41)$$

$$\begin{aligned}
\leq \quad & \frac{1}{2\beta}\left(\|\hat{\boldsymbol{\lambda}}^k-\boldsymbol{\lambda}^*\|^2-\mathbb{E}\left[\|\hat{\boldsymbol{\lambda}}^{k+1}-\boldsymbol{\lambda}^*\|^2\right]\right) \\
& +\frac{\theta_1}{2}\|\frac{\mathbf{x}_1^k-(1-\theta_1-\theta_2)\mathbf{x}_1^{k-1}-\theta_2\tilde{\mathbf{x}}_1}{\theta_1}-\mathbf{x}_1^*\|^2_{\left(L_1+\frac{\|\mathbf{A}_1^T\mathbf{A}_1\|}{\theta_1}\right)\mathbf{I}-\frac{\mathbf{A}_1^T\mathbf{A}_1}{\theta_1}} \\
& -\frac{\theta_1}{2}\mathbb{E}\left(\|\frac{\mathbf{x}_1^{k+1}-(1-\theta_1-\theta_2)\mathbf{x}_1^k-\theta_2\tilde{\mathbf{x}}_1}{\theta_1}-\mathbf{x}_1^*\|^2_{\left(L_1+\frac{\|\mathbf{A}_1^T\mathbf{A}_1\|}{\theta_1}\right)\mathbf{I}-\frac{\mathbf{A}_1^T\mathbf{A}_1}{\theta_1}}\right) \\
& +\frac{\theta_1}{2}\|\frac{\mathbf{x}_2^k-(1-\theta_1-\theta_2)\mathbf{x}_2^{k-1}-\theta_2\tilde{\mathbf{x}}_2}{\theta_1}-\mathbf{x}_2^*\|^2_{\left(\alpha L_2+\frac{\|\mathbf{A}_2^T\mathbf{A}_2\|}{\theta_1}\right)\mathbf{I}} \\
& -\frac{\theta_1}{2}\mathbb{E}\left(\|\frac{\mathbf{x}_2^{k+1}-(1-\theta_1-\theta_2)\mathbf{x}_2^k-\theta_2\tilde{\mathbf{x}}_2}{\theta_1}-\mathbf{x}_2^*\|^2_{\left(\alpha L_2+\frac{\|\mathbf{A}_2^T\mathbf{A}_2\|}{\theta_1}\right)\mathbf{I}}\right),\quad k\geq 1.
\end{aligned}$$

Adding the subscript $s$ and taking expectation on the first $s$ epoches, and

then summing Eq. (41) with $k$ from 0 to $m-1$, we have

$$\frac{1}{\theta_{1,s}}\mathbb{E}\left(L(\mathbf{x}_s^m, \boldsymbol{\lambda}^*) - L(\mathbf{x}^*, \boldsymbol{\lambda}^*)\right) + \frac{\theta_2 + \theta_{1,s}}{\theta_{1,s}}\sum_{k=1}^{m-1}\mathbb{E}\left(L(\mathbf{x}_s^k, \boldsymbol{\lambda}^*) - L(\mathbf{x}^*, \boldsymbol{\lambda}^*)\right)$$

$$\leq \quad \frac{1 - \theta_{1,s} - \theta_2}{\theta_{1,s}}\mathbb{E}\left(L(\mathbf{x}_s^0, \boldsymbol{\lambda}^*) - L(\mathbf{x}^*, \boldsymbol{\lambda}^*)\right) + \frac{m\theta_2}{\theta_{1,s}}\mathbb{E}\left(L(\tilde{\mathbf{x}}_s, \boldsymbol{\lambda}^*) - L(\mathbf{x}^*, \boldsymbol{\lambda}^*)\right)$$

$$+\frac{1}{2}\mathbb{E}\|\frac{\mathbf{y}_{s,1}^0 - \theta_2\tilde{\mathbf{x}}_{s,1} - (1 - \theta_{1,s} - \theta_2)\mathbf{x}_{s,1}^0}{\theta_{1,s}} - \mathbf{x}_1^*\|_{\left(\theta_{1,s}L_1 + \|\mathbf{A}_1^T\mathbf{A}_1\|\right)\mathbf{I} - \mathbf{A}_1^T\mathbf{A}_1}^2$$

$$-\frac{1}{2}\mathbb{E}\|\frac{\mathbf{x}_{s,1}^m - \theta_2\tilde{\mathbf{x}}_{s,1} - (1 - \theta_{1,s} - \theta_2)\mathbf{x}_{s,1}^{m-1}}{\theta_{1,s}} - \mathbf{x}_1^*\|_{\left(\theta_{1,s}L_1 + \|\mathbf{A}_1^T\mathbf{A}_1\|\right)\mathbf{I} - \mathbf{A}_1^T\mathbf{A}_1}^2$$

$$+\frac{1}{2}\mathbb{E}\|\frac{\mathbf{y}_{s,2}^0 - \theta_2\tilde{\mathbf{x}}_{s,2} - (1 - \theta_{1,s} - \theta_2)\mathbf{x}_{s,2}^0}{\theta_{1,s}} - \mathbf{x}_2^*\|_{\left(\alpha\theta_{1,s}L_2 + \|\mathbf{A}_2^T\mathbf{A}_2\|\right)\mathbf{I}}^2$$

$$-\frac{1}{2}\mathbb{E}\|\frac{\mathbf{x}_{s,2}^m - \theta_2\tilde{\mathbf{x}}_{s,2} - (1 - \theta_{1,s} - \theta_2)\mathbf{x}_{s,2}^{m-1}}{\theta_{1,s}} - \mathbf{x}_2^*\|_{\left(\alpha\theta_{1,s}L_2 + \|\mathbf{A}_2^T\mathbf{A}_2\|\right)\mathbf{I}}^2$$

$$+\frac{1}{2\beta}\left(\mathbb{E}\|\hat{\boldsymbol{\lambda}}_s^0 - \boldsymbol{\lambda}^*\|^2 - \mathbb{E}\left[\|\hat{\boldsymbol{\lambda}}_s^m - \boldsymbol{\lambda}^*\|^2\right]\right), \tag{42}$$

where we use $L(\mathbf{x}_s^k, \boldsymbol{\lambda}^*)$ and $L(\tilde{\mathbf{x}}_s, \boldsymbol{\lambda}^*)$ to denote $L(\mathbf{x}_{s,1}^k, \mathbf{x}_{s,2}^k, \boldsymbol{\lambda}^*)$ and $L(\tilde{\mathbf{x}}_{s,1}, \tilde{\mathbf{x}}_{s,2}, \boldsymbol{\lambda}^*)$, respectively. Since $L(\mathbf{x}, \boldsymbol{\lambda}^*)$ is convex for $\mathbf{x}$, we have

$$mL(\tilde{\mathbf{x}}_s, \boldsymbol{\lambda}^*)$$

$$= \quad mL\left(\frac{1}{m}\left(\left[1 - \frac{(\tau-1)\theta_{1,s}}{\theta_2}\right]\mathbf{x}_{s-1}^m + \left[1 + \frac{(\tau-1)\theta_{1,s}}{(m-1)\theta_2}\right]\sum_{k=1}^{m-1}\mathbf{x}_{s-1}^k\right), \boldsymbol{\lambda}^*\right)$$

$$\leq \quad \left[1 - \frac{(\tau-1)\theta_{1,s}}{\theta_2}\right]L(\mathbf{x}_{s-1}^m, \boldsymbol{\lambda}^*) + \left[1 + \frac{(\tau-1)\theta_{1,s}}{(m-1)\theta_2}\right]\sum_{k=1}^{m-1}L(\mathbf{x}_{s-1}^k, \boldsymbol{\lambda}^*), \tag{43}$$

Substituting Eq. (43) into Eq. (42), and using $\mathbf{x}_{s-1}^m = \mathbf{x}_s^0$, we have

$$
\frac{1}{\theta_{1,s}} \mathbb{E}\left(L(\mathbf{x}_s^m, \boldsymbol{\lambda}^*) - L(\mathbf{x}^*, \boldsymbol{\lambda}^*)\right) + \frac{\theta_2 + \theta_{1,s}}{\theta_{1,s}} \sum_{k=1}^{m-1} \mathbb{E}\left(L(\mathbf{x}_s^k, \boldsymbol{\lambda}^*) - L(\mathbf{x}^*, \boldsymbol{\lambda}^*)\right)
$$

$$
\leq \quad \frac{1 - \theta_{1,s} - (\tau-1)\theta_{1,s}}{\theta_{1,s}} \mathbb{E}\left(L(\mathbf{x}_{s-1}^m, \boldsymbol{\lambda}^*) - L(\mathbf{x}^*, \boldsymbol{\lambda}^*)\right)
$$

$$
+ \frac{\theta_2 + \frac{\tau-1}{m-1}\theta_{1,s}}{\theta_{1,s}} \sum_{k=1}^{m-1} \mathbb{E}\left(L(\mathbf{x}_{s-1}^k, \boldsymbol{\lambda}^*) - L(\mathbf{x}^*, \boldsymbol{\lambda}^*)\right)
$$

$$
+ \frac{1}{2}\mathbb{E}\left\| \frac{\mathbf{y}_{s,1}^0 - \theta_2\tilde{\mathbf{x}}_{s,1} - (1 - \theta_{1,s} - \theta_2)\mathbf{x}_{s,1}^0}{\theta_{1,s}} - \mathbf{x}_1^* \right\|_{(\theta_{1,s}L_1 + \|\mathbf{A}_1^T\mathbf{A}_1\|)\mathbf{I} - \mathbf{A}_1^T\mathbf{A}_1}^2
$$

$$
- \frac{1}{2}\mathbb{E}\left\| \frac{\mathbf{x}_{s,1}^m - \theta_2\tilde{\mathbf{x}}_{s,1} - (1 - \theta_{1,s} - \theta_2)\mathbf{x}_{s,1}^{m-1}}{\theta_{1,s}} - \mathbf{x}_1^* \right\|_{(\theta_{1,s}L_1 + \|\mathbf{A}_1^T\mathbf{A}_1\|)\mathbf{I} - \mathbf{A}_1^T\mathbf{A}_1}^2
$$

$$
+ \frac{1}{2}\mathbb{E}\left\| \frac{\mathbf{y}_{s,2}^0 - \theta_2\tilde{\mathbf{x}}_{s,2} - (1 - \theta_{1,s} - \theta_2)\mathbf{x}_{s,2}^0}{\theta_{1,s}} - \mathbf{x}_2^* \right\|_{(\alpha\theta_{1,s}L_2 + \|\mathbf{A}_2^T\mathbf{A}_2\|)\mathbf{I}}^2
$$

$$
- \frac{1}{2}\mathbb{E}\left\| \frac{\mathbf{x}_{s,2}^m - \theta_2\tilde{\mathbf{x}}_{s,2} - (1 - \theta_{1,s} - \theta_2)\mathbf{x}_{s,2}^{m-1}}{\theta_{1,s}} - \mathbf{x}_2^* \right\|_{(\alpha\theta_{1,s}L_2 + \|\mathbf{A}_2^T\mathbf{A}_2\|)\mathbf{I}}^2
$$

$$
+ \frac{1}{2\beta}\left( \mathbb{E}\|\hat{\boldsymbol{\lambda}}_s^0 - \boldsymbol{\lambda}^*\|^2 - \mathbb{E}\left[\|\hat{\boldsymbol{\lambda}}_s^m - \boldsymbol{\lambda}^*\|^2\right] \right). \tag{44}
$$

Then through the setting of $\theta_{1,s} = \frac{1}{2+\tau s}$ and $\theta_2 = \frac{m-\tau}{\tau(m-1)}$, we have

$$
\frac{1}{\theta_{1,s}} = \frac{1 - \tau\theta_{1,s+1}}{\theta_{1,s+1}}, \quad s \geq 0, \tag{45}
$$

and

$$
\frac{\theta_2 + \theta_{1,s}}{\theta_{1,s}} = \frac{\theta_2}{\theta_{1,s+1}} - \tau\theta_2 + 1 = \frac{\theta_2 + \frac{\tau-1}{m-1}\theta_{1,s+1}}{\theta_{1,s+1}}, \quad s \geq 0. \tag{46}
$$

Substituting Eq. (45) into the first term and Eq. (46) into the second term in

the right hand of Eq. (44), we obtain

$$\frac{1}{\theta_{1,s}}\mathbb{E}\left(L(\mathbf{x}_s^m, \boldsymbol{\lambda}^*) - L(\mathbf{x}^*, \boldsymbol{\lambda}^*)\right) + \frac{\theta_2 + \theta_{1,s}}{\theta_{1,s}} \sum_{k=1}^{m-1} \mathbb{E}\left(L(\mathbf{x}_s^k, \boldsymbol{\lambda}^*) - L(\mathbf{x}^*, \boldsymbol{\lambda}^*)\right)$$

$$\leq \quad \frac{1}{\theta_{1,s-1}}\mathbb{E}\left(L(\mathbf{x}_{s-1}^m, \boldsymbol{\lambda}^*) - L(\mathbf{x}^*, \boldsymbol{\lambda}^*)\right) + \frac{\theta_2 + \theta_{1,s-1}}{\theta_{1,s-1}} \sum_{k=1}^{m-1} \mathbb{E}\left(L(\mathbf{x}_{s-1}^k, \boldsymbol{\lambda}^*) - L(\mathbf{x}^*, \boldsymbol{\lambda}^*)\right)$$

$$+\frac{1}{2}\mathbb{E}\|\frac{\mathbf{y}_{s,1}^0 - \theta_2\tilde{\mathbf{x}}_{s,1} - (1-\theta_{1,s}-\theta_2)\mathbf{x}_{s,1}^0}{\theta_{1,s}} - \mathbf{x}_1^*\|_{\left(\theta_{1,s}L_1 + \|\mathbf{A}_1^T\mathbf{A}_1\|\right)\mathbf{I} - \mathbf{A}_1^T\mathbf{A}_1}^2$$

$$-\frac{1}{2}\mathbb{E}\|\frac{\mathbf{x}_{s,1}^m - \theta_2\tilde{\mathbf{x}}_{s,1} - (1-\theta_{1,s}-\theta_2)\mathbf{x}_{s,1}^{m-1}}{\theta_{1,s}} - \mathbf{x}_1^*\|_{\left(\theta_{1,s}L_1 + \|\mathbf{A}_1^T\mathbf{A}_1\|\right)\mathbf{I} - \mathbf{A}_1^T\mathbf{A}_1}^2$$

$$+\frac{1}{2}\mathbb{E}\|\frac{\mathbf{y}_{s,2}^0 - \theta_2\tilde{\mathbf{x}}_{s,2} - (1-\theta_{1,s}-\theta_2)\mathbf{x}_{s,2}^0}{\theta_{1,s}} - \mathbf{x}_2^*\|_{\left(\alpha\theta_{1,s}L_2 + \|\mathbf{A}_2^T\mathbf{A}_2\|\right)\mathbf{I}}^2$$

$$-\frac{1}{2}\mathbb{E}\|\frac{\mathbf{x}_{s,2}^m - \theta_2\tilde{\mathbf{x}}_{s,2} - (1-\theta_{1,s}-\theta_2)\mathbf{x}_{s,2}^{m-1}}{\theta_{1,s}} - \mathbf{x}_2^*\|_{\left(\alpha\theta_{1,s}L_2 + \|\mathbf{A}_2^T\mathbf{A}_2\|\right)\mathbf{I}}^2$$

$$+\frac{1}{2\beta}\left(\mathbb{E}\|\hat{\boldsymbol{\lambda}}_s^0 - \boldsymbol{\lambda}^*\|^2 - \mathbb{E}\left[\|\hat{\boldsymbol{\lambda}}_s^m - \boldsymbol{\lambda}^*\|^2\right]\right). \tag{47}$$

**Proof of Theorem 1**
When $k = 0$, for

$$\mathbf{y}_{s+1}^0 = (1-\theta_2)\mathbf{x}_s^m + \theta_2\tilde{\mathbf{x}}_{s+1} + \frac{\theta_{1,s+1}}{\theta_{1,s}}\left[(1-\theta_{1,s})\mathbf{x}_s^m - (1-\theta_{1,s}-\theta_2)\mathbf{x}_s^{m-1} - \theta_2\tilde{\mathbf{x}}_s\right], \tag{48}$$

we obtain

$$\frac{\mathbf{x}_s^m - \theta_2\tilde{\mathbf{x}}_s - (1-\theta_{1,s}-\theta_2)\mathbf{x}_s^{m-1}}{\theta_{1,s}} = \frac{\mathbf{y}_{s+1}^0 - \theta_2\tilde{\mathbf{x}}_{s+1} - (1-\theta_{1,s+1}-\theta_2)\mathbf{x}_{s+1}^0}{\theta_{1,s+1}}. \tag{49}$$

Substituting Eq. (49) into the third and the fifth terms in the right hand of Eq. (47) and substituting Eq. (30) into the last term in the right hand of Eq. (47),

we obtain

$$\frac{1}{\theta_{1,s}}\mathbb{E}\left(L(\mathbf{x}_s^m,\boldsymbol{\lambda}^*)-L(\mathbf{x}^*,\boldsymbol{\lambda}^*)\right)+\frac{\theta_2+\theta_{1,s}}{\theta_{1,s}}\sum_{k=1}^{m-1}\mathbb{E}\left(L(\mathbf{x}_s^k,\boldsymbol{\lambda}^*)-L(\mathbf{x}^*,\boldsymbol{\lambda}^*)\right)\qquad(50)$$

$$\leq\quad\frac{1}{\theta_{1,s-1}}\mathbb{E}\left(L(\mathbf{x}_{s-1}^m,\boldsymbol{\lambda}^*)-L(\mathbf{x}^*,\boldsymbol{\lambda}^*)\right)+\frac{\theta_2+\theta_{1,s-1}}{\theta_{1,s-1}}\sum_{k=1}^{m-1}\mathbb{E}\left(L(\mathbf{x}_{s-1}^k,\boldsymbol{\lambda}^*)-L(\mathbf{x}^*,\boldsymbol{\lambda}^*)\right)$$

$$+\frac{1}{2}\mathbb{E}\|\frac{\mathbf{x}_{s-1,1}^m-\theta_2\tilde{\mathbf{x}}_{s-1,1}-(1-\theta_{1,s-1}-\theta_2)\mathbf{x}_{s-1,1}^{m-1}}{\theta_{1,s-1}}-\mathbf{x}_1^*\|_{(\theta_{1,s}L_1+\|\mathbf{A}_1^T\mathbf{A}_1\|)\mathbf{I}-\mathbf{A}_1^T\mathbf{A}_1}^2$$

$$-\frac{1}{2}\mathbb{E}\|\frac{\mathbf{x}_{s,1}^m-\theta_2\tilde{\mathbf{x}}_{s,1}-(1-\theta_{1,s}-\theta_2)\mathbf{x}_{s,1}^{m-1}}{\theta_{1,s}}-\mathbf{x}_1^*\|_{(\theta_{1,s}L_1+\|\mathbf{A}_1^T\mathbf{A}_1\|)\mathbf{I}-\mathbf{A}_1^T\mathbf{A}_1}^2$$

$$+\frac{1}{2}\mathbb{E}\|\frac{\mathbf{x}_{s-1,2}^m-\theta_2\tilde{\mathbf{x}}_{s-1,2}-(1-\theta_{1,s-1}-\theta_2)\mathbf{x}_{s-1,2}^{m-1}}{\theta_{1,s-1}}-\mathbf{x}_2^*\|_{(\alpha\theta_{1,s}L_2+\|\mathbf{A}_2^T\mathbf{A}_2\|)\mathbf{I}}^2$$

$$-\frac{1}{2}\mathbb{E}\|\frac{\mathbf{x}_{s,2}^m-\theta_2\tilde{\mathbf{x}}_{s,2}-(1-\theta_{1,s}-\theta_2)\mathbf{x}_{s,2}^{m-1}}{\theta_{1,s}}-\mathbf{x}_2^*\|_{(\alpha\theta_{1,s}L_2+\|\mathbf{A}_2^T\mathbf{A}_2\|)\mathbf{I}}^2$$

$$+\frac{1}{2\beta}\left(\mathbb{E}\|\hat{\boldsymbol{\lambda}}_{s-1}^m-\boldsymbol{\lambda}^*\|^2-\mathbb{E}\left[\|\hat{\boldsymbol{\lambda}}_s^m-\boldsymbol{\lambda}^*\|^2\right]\right),\quad s\geq1.$$

For $\theta_{1,s-1}\geq\theta_{1,s}$, so $\|\mathbf{x}\|_{\theta_{1,s-1}L}^2\geq\|\mathbf{x}\|_{\theta_{1,s}L}^2$, we get

$$\frac{1}{\theta_{1,s}}\mathbb{E}\left(L(\mathbf{x}_s^m,\boldsymbol{\lambda}^*)-L(\mathbf{x}^*,\boldsymbol{\lambda}^*)\right)+\frac{\theta_2+\theta_{1,s}}{\theta_{1,s}}\sum_{k=1}^{m-1}\mathbb{E}\left(L(\mathbf{x}_s^k,\boldsymbol{\lambda}^*)-L(\mathbf{x}^*,\boldsymbol{\lambda}^*)\right)\qquad(51)$$

$$\leq\quad\frac{1}{\theta_{1,s-1}}\mathbb{E}\left(L(\mathbf{x}_{s-1}^m,\boldsymbol{\lambda}^*)-L(\mathbf{x}^*,\boldsymbol{\lambda}^*)\right)+\frac{\theta_2+\theta_{1,s-1}}{\theta_{1,s-1}}\sum_{k=1}^{m-1}\mathbb{E}\left(L(\mathbf{x}_{s-1}^k,\boldsymbol{\lambda}^*)-L(\mathbf{x}^*,\boldsymbol{\lambda}^*)\right)$$

$$+\frac{1}{2}\mathbb{E}\|\frac{\mathbf{x}_{s-1,1}^m-\theta_2\tilde{\mathbf{x}}_{s-1,1}-(1-\theta_{1,s-1}-\theta_2)\mathbf{x}_{s-1,1}^{m-1}}{\theta_{1,s-1}}-\mathbf{x}_1^*\|_{(\theta_{1,s-1}L_1+\|\mathbf{A}_1^T\mathbf{A}_1\|)\mathbf{I}-\mathbf{A}_1^T\mathbf{A}_1}^2$$

$$-\frac{1}{2}\mathbb{E}\|\frac{\mathbf{x}_{s,1}^m-\theta_2\tilde{\mathbf{x}}_{s,1}-(1-\theta_{1,s}-\theta_2)\mathbf{x}_{s,1}^{m-1}}{\theta_{1,s}}-\mathbf{x}_1^*\|_{(\theta_{1,s}L_1+\|\mathbf{A}_1^T\mathbf{A}_1\|)\mathbf{I}-\mathbf{A}_1^T\mathbf{A}_1}^2$$

$$+\frac{1}{2}\mathbb{E}\|\frac{\mathbf{x}_{s-1,2}^m-\theta_2\tilde{\mathbf{x}}_{s-1,2}-(1-\theta_{1,s-1}-\theta_2)\mathbf{x}_{s-1,2}^{m-1}}{\theta_{1,s-1}}-\mathbf{x}_2^*\|_{(\alpha\theta_{1,s-1}L_2+\|\mathbf{A}_2^T\mathbf{A}_2\|)\mathbf{I}}^2$$

$$-\frac{1}{2}\mathbb{E}\|\frac{\mathbf{x}_{s,2}^m-\theta_2\tilde{\mathbf{x}}_{s,2}-(1-\theta_{1,s}-\theta_2)\mathbf{x}_{s,2}^{m-1}}{\theta_{1,s}}-\mathbf{x}_2^*\|_{(\alpha\theta_{1,s}L_2+\|\mathbf{A}_2^T\mathbf{A}_2\|)\mathbf{I}}^2$$

$$+\frac{1}{2\beta}\left(\mathbb{E}\|\hat{\boldsymbol{\lambda}}_{s-1}^m-\boldsymbol{\lambda}^*\|^2-\mathbb{E}\left[\|\hat{\boldsymbol{\lambda}}_s^m-\boldsymbol{\lambda}^*\|^2\right]\right),\quad s\geq1,$$

When $s=0$, through Eq. (47), and using that $\mathbf{y}_{0,1}^0=\tilde{\mathbf{x}}_{0,1}=\mathbf{x}_{0,1}^0$ and

$\mathbf{y}_{0,2}^0 = \tilde{\mathbf{x}}_{0,2} = \mathbf{x}_{0,2}^0$, we obtain

$$\frac{1}{\theta_{1,0}}\mathbb{E}\left(L(\mathbf{x}_0^m, \boldsymbol{\lambda}^*)\right) - L(\mathbf{x}^*, \boldsymbol{\lambda}^*)) + \frac{\theta_{1,0} + \theta_2}{\theta_{1,0}}\sum_{k=1}^{m-1}\mathbb{E}\left(L(\mathbf{x}_0^k, \boldsymbol{\lambda}^*) - L(\mathbf{x}^*, \boldsymbol{\lambda}^*)\right)$$

$$\leq \quad \frac{1 - \theta_{1,0} + (m-1)\theta_2}{\theta_{1,0}}\left(L(\mathbf{x}_0, \boldsymbol{\lambda}^*)\right) - L(\mathbf{x}^*, \boldsymbol{\lambda}^*))$$

$$+ \frac{1}{2}\|\mathbf{x}_{0,1}^0 - \mathbf{x}_1^*\|_{\left(\theta_{1,0}L_1 + \|\mathbf{A}_1^T\mathbf{A}_1\|\right)\mathbf{I} - \mathbf{A}_1^T\mathbf{A}_1}^2$$

$$- \frac{1}{2}\mathbb{E}\|\frac{\mathbf{x}_{0,1}^m - \theta_2\tilde{\mathbf{x}}_{0,1} - (1 - \theta_{1,0} - \theta_2)\mathbf{x}_{0,1}^{m-1}}{\theta_{1,s=0}} - \mathbf{x}_1^*\|_{\left(\theta_{1,0}L_1 + \|\mathbf{A}_1^T\mathbf{A}_1\|\right)\mathbf{I} - \mathbf{A}_1^T\mathbf{A}_1}^2$$

$$+ \frac{1}{2}\|\mathbf{x}_{0,2}^0 - \mathbf{x}_1^*\|_{\left(\alpha\theta_{1,0}L_2 + \|\mathbf{A}_2^T\mathbf{A}_2\|\right)\mathbf{I}}^2$$

$$- \frac{1}{2}\mathbb{E}\|\frac{\mathbf{x}_{0,2}^m - \theta_2\tilde{\mathbf{x}}_{0,2} - (1 - \theta_{1,0} - \theta_2)\mathbf{x}_{0,2}^{m-1}}{\theta_{1,0}} - \mathbf{x}_2^*\|_{\left(\alpha\theta_{1,0}L_2 + \|\mathbf{A}_2^T\mathbf{A}_2\|\right)\mathbf{I}}^2$$

$$+ \frac{1}{2\beta}\left(\|\hat{\boldsymbol{\lambda}}_0^0 - \boldsymbol{\lambda}^*\|^2 - \mathbb{E}\left[\|\hat{\boldsymbol{\lambda}}_0^m - \boldsymbol{\lambda}^*\|^2\right]\right). \tag{52}$$

Summing Eq. (51) $s$ from 1 to $S-1$ and adding Eq. (52), we have the result that

$$\frac{1}{\theta_{1,S}}\mathbb{E}\left(L(\mathbf{x}_S^m, \boldsymbol{\lambda}^*)\right) - L(\mathbf{x}^*, \boldsymbol{\lambda}^*)) + \frac{\theta_{1,S} + \theta_2}{\theta_{1,S}}\sum_{k=1}^{m-1}\mathbb{E}\left(L(\mathbf{x}_S^k, \boldsymbol{\lambda}^*) - L(\mathbf{x}^*, \boldsymbol{\lambda}^*)\right)$$

$$\leq \quad \frac{1 - \theta_{1,0} + (m-1)\theta_2}{\theta_{1,0}}\left(L(\mathbf{x}_0^0, \boldsymbol{\lambda}^*)\right) - L(\mathbf{x}^*, \boldsymbol{\lambda}^*))$$

$$+ \frac{1}{2}\|\mathbf{x}_{0,1}^0 - \mathbf{x}_1^*\|_{\left(\theta_{1,0}L_1 + \|\mathbf{A}_1^T\mathbf{A}_1\|\right)\mathbf{I} - \mathbf{A}_1^T\mathbf{A}_1}^2 + \frac{1}{2}\|\mathbf{x}_{0,2}^0 - \mathbf{x}_2^*\|_{\left(\alpha\theta_{1,0}L_2 + \|\mathbf{A}_2^T\mathbf{A}_2\|\right)\mathbf{I}}^2$$

$$+ \frac{1}{2\beta}\left(\|\hat{\boldsymbol{\lambda}}_0^0 - \boldsymbol{\lambda}^*\|^2 - \mathbb{E}\left[\|\hat{\boldsymbol{\lambda}}_S^m - \boldsymbol{\lambda}^*\|^2\right]\right)$$

$$- \frac{1}{2}\mathbb{E}\|\frac{\mathbf{x}_{S,1}^m - \theta_2\tilde{\mathbf{x}}_{S,1} - (1 - \theta_{1,s} - \theta_2)\mathbf{x}_{S,1}^{m-1}}{\theta_{1,S}} - \mathbf{x}_1^*\|_{\left(\theta_{1,S}L_1 + \|\mathbf{A}_1^T\mathbf{A}_1\|\right)\mathbf{I} - \mathbf{A}_1^T\mathbf{A}_1}^2$$

$$- \frac{1}{2}\mathbb{E}\|\frac{\mathbf{x}_{S,2}^m - \theta_2\tilde{\mathbf{x}}_{S,2} - (1 - \theta_{1,S} - \theta_2)\mathbf{x}_{s,2}^{m-1}}{\theta_{1,S}} - \mathbf{x}_2^*\|_{\left(\alpha\theta_{1,S}L_2 + \|\mathbf{A}_2^T\mathbf{A}_2\|\right)\mathbf{I}}^2$$

$$\leq \quad \frac{1 - \theta_{1,0} + (m-1)\theta_2}{\theta_{1,0}}\left(L(\mathbf{x}_0^0, \boldsymbol{\lambda}^*)\right) - L(\mathbf{x}^*, \boldsymbol{\lambda}^*))$$

$$+ \frac{1}{2}\|\mathbf{x}_{0,1}^0 - \mathbf{x}_1^*\|_{\left(\theta_{1,0}L_1 + \|\mathbf{A}_1^T\mathbf{A}_1\|\right)\mathbf{I} - \mathbf{A}_1^T\mathbf{A}_1}^2 + \frac{1}{2}\|\mathbf{x}_{0,2}^0 - \mathbf{x}_2^*\|_{\left(\alpha\theta_{1,0}L_2 + \|\mathbf{A}_2^T\mathbf{A}_2\|\right)\mathbf{I}}^2$$

$$+ \frac{1}{2\beta}\left(\|\hat{\boldsymbol{\lambda}}_0^0 - \boldsymbol{\lambda}^*\|^2 - \mathbb{E}\left[\|\hat{\boldsymbol{\lambda}}_S^m - \boldsymbol{\lambda}^*\|^2\right]\right). \tag{53}$$

Now we analyse $\|\hat{\boldsymbol{\lambda}}_S^m - \boldsymbol{\lambda}^*\|^2$. From Eq. (30), for $s \geq 1$, we have

$$\hat{\boldsymbol{\lambda}}_s^m - \hat{\boldsymbol{\lambda}}_{s-1}^m = \hat{\boldsymbol{\lambda}}_s^m - \hat{\boldsymbol{\lambda}}_s^0 = \sum_{k=1}^m \left( \hat{\boldsymbol{\lambda}}_s^k - \hat{\boldsymbol{\lambda}}_s^{k-1} \right)$$

$$\overset{a}{=} \beta \sum_{k=1}^m \left( \frac{1}{\theta_{1,s}} \left( \mathbf{A}\mathbf{x}_s^k - \mathbf{b} \right) - \frac{1 - \theta_{1,s} - \theta_2}{\theta_{1,s}} \left( \mathbf{A}\mathbf{x}_s^{k-1} - \mathbf{b} \right) - \frac{\theta_2}{\theta_{1,s}} \left( \mathbf{A}\tilde{\mathbf{x}}_s - \mathbf{b} \right) \right)$$

$$\overset{b}{=} \frac{\beta}{\theta_{1,s}} \left( \mathbf{A}\mathbf{x}_s^m - \mathbf{b} \right) + \frac{\beta(\theta_2 + \theta_{1,s})}{\theta_{1,s}} \sum_{k=1}^{m-1} \left( \mathbf{A}\mathbf{x}_s^k - \mathbf{b} \right)$$
$$- \frac{\beta(1 - \theta_{1,s} - \theta_2)}{\theta_{1,s}} \left( \mathbf{A}\mathbf{x}_{s-1}^m - \mathbf{b} \right) - \frac{m\beta\theta_2}{\theta_{1,s}} \left( \mathbf{A}\tilde{\mathbf{x}}_{s-1} - \mathbf{b} \right)$$

$$\overset{c}{=} \frac{\beta}{\theta_{1,s}} \left( \mathbf{A}\mathbf{x}_s^m - \mathbf{b} \right) + \frac{\beta(\theta_2 + \theta_{1,s})}{\theta_{1,s}} \sum_{k=1}^{m-1} \left( \mathbf{A}\mathbf{x}_s^k - \mathbf{b} \right)$$
$$- \beta \left( \frac{1 - \theta_{1,s} - (\tau - 1)\theta_{1,s}}{\theta_{1,s}} \left( \mathbf{A}\mathbf{x}_{s-1}^m - \mathbf{b} \right) + \frac{\theta_2 + \frac{\tau}{m-1}\theta_{1,s}}{\theta_{1,s}} \sum_{k=1}^{m-1} \left( \mathbf{A}\mathbf{x}_{s-1}^k - \mathbf{b} \right) \right)$$

$$\overset{d}{=} \frac{\beta}{\theta_{1,s}} \left( \mathbf{A}\mathbf{x}_s^m - \mathbf{b} \right) + \frac{\beta(\theta_2 + \theta_{1,s})}{\theta_{1,s}} \sum_{k=1}^{m-1} \left( \mathbf{A}\mathbf{x}_s^k - \mathbf{b} \right)$$
$$- \frac{\beta}{\theta_{1,s-1}} \left( \mathbf{A}\mathbf{x}_{s-1}^m - \mathbf{b} \right) - \frac{\beta(\theta_2 + \theta_{1,s-1})}{\theta_{1,s-1}} \sum_{k=1}^{m-1} \left( \mathbf{A}\mathbf{x}_{s-1}^k - \mathbf{b} \right), \tag{54}$$

where the equality $\overset{a}{=}$ uses Eq. (28); the equalities $\overset{b}{=}, \overset{c}{=}$, and $\overset{d}{=}$ are obtained through the same techniques of Eq. (42), Eq. (44) and Eq. (47). When $s = 0$, we can obtain

$$\hat{\boldsymbol{\lambda}}_0^m - \hat{\boldsymbol{\lambda}}_0^0 = \sum_{k=1}^m \left( \hat{\boldsymbol{\lambda}}_0^k - \hat{\boldsymbol{\lambda}}_0^{k-1} \right) \tag{55}$$

$$= \sum_{k=1}^m \left( \frac{\beta}{\theta_{1,0}} \left( \mathbf{A}\mathbf{x}_0^k - \mathbf{b} \right) - \frac{\beta(1 - \theta_{1,0} - \theta_2)}{\theta_{1,0}} \left( \mathbf{A}\mathbf{x}_0^{k-1} - \mathbf{b} \right) - \frac{\theta_2 \beta}{\theta_{1,0}} \left( \mathbf{A}\mathbf{x}_0^0 - \mathbf{b} \right) \right)$$

$$= \frac{\beta}{\theta_{1,0}} \left( \mathbf{A}\mathbf{x}_0^m - \mathbf{b} \right) + \frac{\beta(\theta_2 + \theta_{1,0})}{\theta_{1,0}} \sum_{k=1}^{m-1} \left( \mathbf{A}\mathbf{x}_0^k - \mathbf{b} \right) - \frac{\beta(1 - \theta_{1,0} + (m-1)\theta_2)}{\theta_{1,0}} \left( \mathbf{A}\mathbf{x}_0^0 - \mathbf{b} \right).$$

Summing Eq. (54) with $s$ from 1 to $S - 1$ and adding Eq. (55), we have the

result that

$$
\hat{\boldsymbol{\lambda}}_S^m - \boldsymbol{\lambda}^* = \hat{\boldsymbol{\lambda}}_S^m - \hat{\boldsymbol{\lambda}}_0^0 + \hat{\boldsymbol{\lambda}}_0^0 - \boldsymbol{\lambda}^*
$$

$$
= \frac{\beta}{\theta_{1,S}}\left(\mathbf{A}\mathbf{x}_S^m - \mathbf{b}\right) + \frac{\beta(\theta_2 + \theta_{1,S})}{\theta_{1,S}}\sum_{k=1}^{m-1}\left(\mathbf{A}\mathbf{x}_S^k - \mathbf{b}\right) - \frac{\beta\left(1 - \theta_{1,0} + (m-1)\theta_2\right)}{\theta_{1,0}}\left(\mathbf{A}\mathbf{x}_0^0 - \mathbf{b}\right)
$$

$$
+ \tilde{\boldsymbol{\lambda}}_0^0 + \frac{\beta(1 - \theta_{1,0})}{\theta_{1,0}}\left(\mathbf{A}\mathbf{x}_0^0 - \mathbf{b}\right) - \boldsymbol{\lambda}^*
$$

$$
\stackrel{a}{=} \frac{m\beta}{\theta_{1,S}}\left(\mathbf{A}\hat{\mathbf{x}}_S - \mathbf{b}\right) + \tilde{\boldsymbol{\lambda}}_0^0 - \frac{\beta(m-1)\theta_2}{\theta_{1,0}}\left(\mathbf{A}\mathbf{x}_0^0 - \mathbf{b}\right) - \boldsymbol{\lambda}^*. \tag{56}
$$

where ithe equality $\stackrel{a}{=}$ uses the definition of $\hat{\mathbf{x}}_S$. Substituting Eq. (56) into Eq. (53), we can obtain Theorem 1.

**Proof of Corollary 1**
We set

$$
C_1 = \frac{1 - \theta_{1,0} + (m-1)\theta_2}{\theta_{1,0}}\left(F(\mathbf{x}_0^0) - F(\mathbf{x}^*) + \langle \boldsymbol{\lambda}^*, \mathbf{A}\mathbf{x}_0^0 - \mathbf{b}\rangle\right) \tag{57}
$$

$$
+ \frac{1}{2\beta}\|\tilde{\boldsymbol{\lambda}}_0^0 + \frac{\beta(1 - \theta_{1,0})}{\theta_{1,0}}(\mathbf{A}\mathbf{x}_0^0 - \mathbf{b}) - \boldsymbol{\lambda}^*\|^2
$$

$$
+ \frac{1}{2}\|\mathbf{x}_{0,1}^0 - \mathbf{x}_1^*\|^2_{\left(\theta_{1,0}L_1 + \|\mathbf{A}_1^T\mathbf{A}_1\|\right)\mathbf{I} - \mathbf{A}_1^T\mathbf{A}_1} + \frac{1}{2}\|\mathbf{x}_{0,2}^0 - \mathbf{x}_2^*\|^2_{\left(\left(1 + \frac{1}{b\theta_2}\right)\theta_{1,0}L_2\right)\mathbf{I} + \|\mathbf{A}_2^T\mathbf{A}_2\|}.
$$

Since $F(\mathbf{x})$ is convex,

$$
F(\hat{\mathbf{x}}_S) - F(\mathbf{x}^*) + \langle \boldsymbol{\lambda}^*, \mathbf{A}\hat{\mathbf{x}}_S - \mathbf{b}\rangle \geq 0.
$$

Taking expectation, we obtain:

$$
\mathbb{E}\left(F(\hat{\mathbf{x}}_S) - F(\mathbf{x}^*) + \langle \boldsymbol{\lambda}^*, \mathbf{A}\hat{\mathbf{x}}_S - \mathbf{b}\rangle\right) \geq 0.
$$

Then from Theorem 1, we obtain

$$
\mathbb{E}\left(F(\hat{\mathbf{x}}_S) - F(\mathbf{x}^*) + \langle \boldsymbol{\lambda}^*, \mathbf{A}\hat{\mathbf{x}}_S - \mathbf{b}\rangle\right) \leq \frac{C_1}{m}\theta_{1,S}, \tag{58}
$$

and

$$
\mathbb{E}\|\frac{m\beta}{\theta_{1,S}}\left(\mathbf{A}\hat{\mathbf{x}}_S - \mathbf{b}\right) + \boldsymbol{\lambda}_0^0 - \frac{\beta(m-1)\theta_2}{\theta_{1,0}}\left(\mathbf{A}\mathbf{x}_0 - \mathbf{b}\right) - \boldsymbol{\lambda}^*\|^2 \leq 2\beta C_1, \tag{59}
$$

So

$$
\mathbb{E}\|\frac{m\beta}{\theta_{1,S}}\left(\mathbf{A}\hat{\mathbf{x}}_S - \mathbf{b}\right) + \boldsymbol{\lambda}_0^0 - \frac{\beta(m-1)\theta_2}{\theta_{1,0}}\left(\mathbf{A}\mathbf{x}_0 - \mathbf{b}\right) - \boldsymbol{\lambda}^*\| \leq \sqrt{2\beta C_1}, \tag{60}
$$

where we use the fact that $0 \leq \mathbb{E}\left(\xi - \mathbb{E}(\xi)\right)^2 = \mathbb{E}|\xi|^2 - |\mathbb{E}\xi|^2$, and set $\xi = \|\frac{m\beta}{\theta_{1,S}}\left(\mathbf{A}\hat{\mathbf{x}}_S - \mathbf{b}\right) + \boldsymbol{\lambda}_0^0 - \frac{\beta(m-1)\theta_2}{\theta_{1,0}}\left(\mathbf{A}\mathbf{x}_0 - \mathbf{b}\right) - \boldsymbol{\lambda}^*\|$. Since $\|\mathbf{a} - \mathbf{b}\| \geq \|\mathbf{a}\| - \|\mathbf{b}\|$, we obtain

$$
\mathbb{E}\|\frac{m\beta}{\theta_{1,S}}\left(\mathbf{A}\hat{\mathbf{x}}_S - \mathbf{b}\right)\| \leq C_2, \tag{61}
$$

where $C_2 = \sqrt{2\beta C_1} + \|\boldsymbol{\lambda}_0^0 - \frac{\beta(m-1)\theta_2}{\theta_{1,0}}(\mathbf{A}\mathbf{x}_0 - \mathbf{b}) - \boldsymbol{\lambda}^*\|$. Thus

$$\mathbb{E}\|\mathbf{A}\hat{\mathbf{x}}_S - \mathbf{b}\| \leq \frac{C_2}{m\beta}\theta_{1,S} = O(\frac{1}{S}). \tag{62}$$

For $\mathbb{E}\left(F(\hat{\mathbf{x}}_S) - F(\mathbf{x}^*) + \langle \boldsymbol{\lambda}^*, \mathbf{A}\hat{\mathbf{x}}_S - \mathbf{b}\rangle\right) \geq 0$, we obtain

$$-\mathbb{E}\|\boldsymbol{\lambda}^*\|\|\mathbf{A}\hat{\mathbf{x}}_S - \mathbf{b}\| \leq \mathbb{E}\left(F(\hat{\mathbf{x}}_S) - F(\mathbf{x}^*)\right) \leq \frac{C_1}{m}\theta_{1,S} + \mathbb{E}\|\boldsymbol{\lambda}^*\|\|\mathbf{A}\hat{\mathbf{x}}_S - \mathbf{b}\|. \tag{63}$$

So

$$\mathbb{E}|F(\hat{\mathbf{x}}_S) - F(\mathbf{x}^*)| \leq O(\frac{1}{S}). \tag{64}$$

This ends the proof.

# 3   Experiments

## 3.1   Lasso Problems

We compare our method with (1) STOC-ADMM [5], (2) SVRG-ADMM [8], (3) OPT-SADMM [3], (4) SAG-ADMM [9]. We implement those algorithms as follows:

- STOC-ADMM [5]. The step size for STOC-ADMM $\gamma = 1/(L_2 + \sigma k^{\frac{1}{2}} + \beta\|\mathbf{A}^T\mathbf{A}\|)$. We set $\beta_s = \min(10, \rho^s \beta_0)$ and tune $\sigma$ from $\{10^{-5}, 10^{-4}, 10^{-3}\}$.

- OPT-ADMM [3]. The step size for OPT-ADMM $\gamma = 1/(L_2 + \sigma k^{\frac{3}{2}} + \beta\|\mathbf{A}^T\mathbf{A}\|)$. We set $\beta_s = \min(10, \rho^s \beta_0)$ and tune $\sigma$ from $\{10^{-7}, 10^{-6}, 10^{-5}\}$.

- SVRG-ADMM [8]. The step size for SVRG-ADMM $\gamma = 1/(L_2 + \beta\|\mathbf{A}^T\mathbf{A}\|)$. We set $\beta_s = \min(10, \rho^s \beta_0)$.

- SAG-ADMM [9]. The step size for SAG-ADMM $\gamma = 1/(L_2 + \beta)$. We set $\beta_s = \min(10, \rho^s \beta_0)$.

- ACC-SADMM (ours). The step size for ACC-SADMM is $\gamma = 1/(L_2(1 + \frac{2}{b}) + \frac{\beta_0}{\theta_{1,s}}\|\mathbf{A}^T\mathbf{A}\|)$.

For all the other algorithms, we tune $\rho$ from $\{1, 1.05, 1.1, 1.3\}$. And we tune $\beta_0$ from $\{10^{-4}, 10^{-3}, 10^{-2}, 10^{-1}\}$. We normalize the Frobenius norm of each feature to 1. For the original Lasso problem, $L_2 = 1$. For the Graph-Guided Fused Lasso problem, $L_2$ is tuned from $\{1 \times 10^k, 2 \times 10^k, 5 \times 10^k | -5 \leq k \leq -1, k \in \mathcal{Z}\}$ to obtain the best step size for each algorithm.

In experiment, we first fix $\sigma = 0$ and $\rho = 1$ and then tune the parameters $\beta_0$ and $L_2$ based on the first 10 data passes. Then we retune the parameters for $\sigma$ and $\rho$. For some algorithms, there are 4 parameters to tune. However, we find that the major factors of the speed for the algorithms are $\beta_0$ and $L_2$.

Fig. 1 shows more experimental results with fixed $L_2 = 0.01$ for the original Lasso problem and the Graph-Guided Fused Lasso problem on the a9a and mnist datasets. Fig. 2 and 3 reports the testing loss for original Lasso and Graph-Guided Fused Lasso. Table 1 reports the memory costs of all algorithms.

(a) a9a-original Lasso         (b) mnist-original Lasso

(c) a9a-Graph-Guided Lasso     (d) mnist-Graph-Guided Lasso

Figure 1: Experimental results of solving the original Lasso and the Graph-Guided Fused Lasso problem on the a9a and mnist datasets with $L_2 = 0.01$.

Table 1: Memory Costs for Storing Data on Different Datasets.

|            | a9a     | covertype | mnist  | dna     | ImageNet |
|------------|---------|-----------|--------|---------|----------|
| STOC-ADMM  | 2.31KB  | 1.69KB    | 123KB  | 25.0KB  | 62.5MB   |
| OPT-ADMM   | 2.89KB  | 2.10KB    | 153KB  | 31.3KB  | 78.1MB   |
| SVRG-ADMM  | 3.47KB  | 2.53KB    | 184KB  | 37.5KB  | 93.8MB   |
| SAG-ADMM   | 82.9MB  | 0.23GB    | 3.50GB | 28.6GB  | 38.2TB   |
| ACC-ADMM   | 7.51MB  | 5.48KB    | 398KB  | 81.3KB  | 208MB    |

## 3.2 Multitask Learning

We perform experiments on multitask learning [2]. A similar experiment is also conducted by [8]. The experiment is performed on a 1000-class ImageNet

(a) a9a-original Lasso

(b) covertype-original Lasso

(c) mnist-original Lasso

(d) dna-original Lasso

STOC-ADMM · STOC-ADMM-ERG · OPT-ADMM · SVRG-ADMM · SVRG-ADMM-ERG · SAG-ADMM · SAG-ADMM-ERG · ACC-SADMM

Figure 2: The curves of testing loss in solving the original Lasso problem corresponding to the experiment in the paper. "-ERG" represents the ergodic results for the corresponding algorithms.

dataset [6]. The features are generated from the last fully connected layer of the convolutional VGG-16 net [7]. Since there is no parameter tuning issue, we use the validation set of ImageNet as the test set of the algorithms being compared. There are $1,281,167$ training images and the validation set includes $50,000$ images. 4096 features are generated from the last fully connected layer of the convolutional VGG-16 net [7]. We solve the problem: $\min_{\mathbf{X}} l(X) + \mu_1 \|\mathbf{X}\|_1 + \mu_2 \|\mathbf{X}\|_*$, where $l(\mathbf{X})$ is the logistic loss. Like [8], we set $\mu_1 = 10^{-4}$ and $\mu_2 = 10^{-5}$. We set the mini-batchsize $b = 2000$ since $\|\mathbf{X}\|_*$ should be solved through Singular Value Decomposition at each step.

Fig. 4 shows the objective gap and test error against iteration. Our method is also faster than other SADMM. Our final test error is 30.9% while using the weight from the softmax layer of the original VGG model [7], the test error is 32.4%.

(a) a9a-Graph-Guided Lasso

(b) covertype-Graph-Guided Lasso

(c) mnist-Graph-Guided Lasso

(d) dna-Graph-Guided Lasso

Figure 3: The curves of testing loss in solving the Graph-Guided Fused Lasso problem corresponding to the experiment in the paper. "-ERG" represents the ergodic results for the corresponding algorithms.

(a) objective gap vs. iteration

(b) test error vs. iteration

Figure 4: The experimental result of Multitask Learning.