[Reviews · NeurIPS 2017]

Reviewer 1



The paper considers the acceleration of stochastic ADMM. The paper basically combines existing techniques like Nesterov's acceleration technique, and increasing penalty parameter and variance reduction technique. The clearness and readability of the paper needs to be improved. The algorithm is designed for purpose of the theoretical bounds but not for a practically faster algorithm. Overall, the paper needs to be improved. 1. In the abstract and introduction, the paper does not clearly point out the result is based on the average of the last m iterations when state non-ergodic convergence rate. The paper also does not point out the use of increasing penalty parameter which is an important trick to establish the theoretical result. 2. Algorithm 2 looks like a warm start algorithm by increasing the penalty parameter (viz. step size). For a warm start algorithm, the theoretical result is expected for the average of the last loop iterates and outer-loop iterates. 3. If the theoretical result could be established without increasing penalty parameter, it is desirable. Trading penalty parameter (speed) for theoretical bound is not a significant contribution.

Reviewer 2



This paper proposes Accelerated Stochastic ADMM(ACC-SADMM) for large scale general convex finite-sum problems with linear functions. In the inner loop, they do extrapolation and update primal and dual variables which ensures they can obtain a non-ergodic results and also accelerate the process and reduce the variance. For outer loop, they preserve snapshot vectors and resets the initial value. The authors may describe the update rule in a more intuitive way since updating formula is a little complicated. The main contribution is that this algorithm has a better convergence rate.

Reviewer 3



Summary: This paper develops a sophisticated stochastic ADMM method that integrates Nesterov acceleration and variance reduction techniques so as to obtain an optimal non-ergodic O(1/K) convergence rate. The experimental results appear to be very compelling. Some of the choices in the design of the algorithm appear to be unintuitive (Algorithm 1 and 2). What is the basis of choosing c=tau=2? How does the penalty factor adjustment proposed in the paper compare with other "adaptive-rho" heuristics such as based on balancing primal/dual tolerances. Are there any differences among stochastic ADMM variants in terms of ability to parallelize? Section 3.3: "we preserves" --> "we preserve"